# Predisaster predictors of posttraumatic stress symptom trajectories: An analysis of low-income women in the aftermath of Hurricane Katrina

**Sarah R. Lowe**[1]*, **Ethan J. Raker**[2], **Mary C. Waters**[2], **Jean E. Rhodes**[3]

**1** Department of Social and Behavioral Sciences, Yale School of Public Health, New Haven, CT, United States of America, **2** Department of Sociology, Harvard University, Cambridge, MA, United States of America, **3** Department of Psychology, University of Massachusetts Boston, Boston, MA, United States of America

\* sarah.lowe@yale.edu

**Data Availability Statement:** The data underlying the results presented in the study are available from the Harvard Dataverse Respository at https://

## Abstract

Exposure to disasters is associated with a range of posttraumatic stress symptom (PTSS) trajectories. However, few studies have tracked PTSS for more than a decade postdisaster, and none to our knowledge has explored the role of predisaster resources and vulnerabilities in shaping the likelihood of trajectory membership. The current study included participants from the Resilience in Survivors of Katrina Study (*N* = 885). Participants were originally part of a community college intervention study and were assessed prior to the hurricane (6–21 months predisaster), and approximately 1 year, 4 years, and 12 years postdisaster. Latent class growth analysis identified PTSS trajectories. Perceived social support, probable mental illness, and physical health conditions or problems–all assessed predisaster–were examined as predictors of trajectory membership at the univariate level and in multivariable models without and with adjustment for disaster exposure. Three PTSS trajectories were detected: *Moderate-Decreasing* (69.3%), *High-Decreasing* (23.1%), and *High-Stable* (7.6%). In the multivariable predictive model without adjustment for disaster exposure, probable predisaster mental illness was significantly associated with greater odds of the *High-Decreasing* and *High-Stable* trajectories, and physical health conditions or problems with greater odds of the *High-Decreasing* trajectory, relative to the *Moderate-Decreasing* trajectory. However, when disaster exposure was adjusted, only the association between predisaster mental illness and the odds of the *High-Stable* trajectory remained statistically significant. Lower predisaster perceived social support was significantly associated with membership in the *High-Decreasing* trajectory, relative to the *Moderate-Decreasing*, at the univariate level, but not in either multivariable model. Whereas predisaster mental illness confers risk for chronic postdisaster PTSS, other impacts of predisaster resources and vulnerabilities on elevated PTSS trajectories do not go beyond those of disaster exposure. The results support disaster preparedness efforts targeting those with mental and physical health conditions, and postdisaster mental health services addressing preexisting vulnerabilities in addition to disaster-related PTSS.

dataverse.harvard.edu/dataset.xhtml?persistentId=
doi:10.7910/DVN/MVM3W0.

**Funding:** This research was supported by the
Eunice Kennedy Shriver National Institute of Child
Health and Human Development (https://www.
nichd.nih.gov/) in the form of grants awarded to
MCW (P01HD082032, R01HD057599,
R01HD046162), the National Science Foundation
(https://www.nsf.gov/) in the form of a grant
awarded to MCW (BCS-0555240), the MacArthur
Foundation (https://www.macfound.org/) in the
form of a grant awarded to MCW (04-80775-000-
HCD), the Robert Wood Johnson Foundation
(https://www.rwjf.org) in the form of a
grant awarded to MCW (23029), the Center for
Economic Policy Studies at Princeton University
(https://gceps.princeton.edu/) in the form of
funding awarded to MCW, the Harvard Center for
Population and Development Studies (https://www.
hsph.harvard.edu/population-development/) in
form of funding awarded to MCW, and an Early-
Career Research Fellowship from the Gulf
Research Program of the National Academies of
Sciences, Engineering, and Medicine (https://www.
nationalacademies.org/) awarded to SL. The
funders had no role in study design, data collection
and analysis, decision to publish, or preparation of
the manuscript. The content is solely the
responsibility of the authors and does not
necessarily represent the official views of the Gulf
Research Program of the National Academies of
Sciences, Engineering, and Medicine.

**Competing interests:** The authors have declared
that no competing interests exist.

## Introduction

A large body of research has shown that natural disasters, including hurricanes, floods, and earthquakes, are associated with a range of adverse mental health outcomes [1–3]. The most commonly explored postdisaster mental health outcome is posttraumatic stress disorder (PTSD), which is characterized by intrusive symptoms, avoidance, negative alterations in cognition and mood, and alterations in arousal and reactivity [4]. The prevalence of postdisaster PTSD after disasters has been estimated at 5–10% in the general population [2]. Among the risk factors for PTSD after disasters are female gender, low socioeconomic status, and exposure to a greater number of disaster-related stressors and traumatic events [2].

Although exploring PTSD and its predictors cross-sectionally is useful for the planning and provision of postdisaster services, scholars have increasingly recognized the limitations of this approach. First, a focus on PTSD as a dichotomous outcome neglects the heterogeneity in posttraumatic stress symptoms (PTSS) among those who do and do not meet criteria for the disorder, such that those with subclinical symptoms might benefit from mental health services, and those with extreme symptoms might have different needs from those who just surpass the diagnostic threshold. Second, cross-sectional research fails to capture varied patterns, or *trajectories*, of PTSS, such as consistently low PTSS (often termed *resilience*), initially high PTSS that decreases rapidly or gradually over time, consistently high PTSS, and initially low PTSS that increases months or years after exposure [5, 6]. Although information on earlier levels of PTSS could be collected at a single time point, this would be inherently subject to retrospective bias. Longitudinal data, with multiple assessments of PTSS over time, overcomes this limitation and thus allows for more precise information on the prevalence and predictors of PTSS trajectories. Such findings are useful for forecasting the proportion and characteristics of survivors who might have service needs in the short- and longer-term aftermath of disasters.

Several studies have explored trajectories of PTSS and other mental health symptoms after disasters and other traumatic events. A recent review identified 67 trajectory analyses across 54 studies, seven of which were conducted in the aftermath of natural disasters and 38 of which included PTSS as a primary outcome [7]. The most commonly observed trajectory was characterized by consistently low symptoms (termed *resilience*); an average of 65.7% of participants exhibited this trajectory across the studies. Trajectories of high-decreasing symptoms (termed *recovery*) and consistently high symptoms (termed *chronic*) were also commonly observed albeit at lower proportions (20.8% and 10.6%, respectively), whereas other patterns, including low-increasing and moderate-increasing symptoms, were less frequent.

One noteworthy finding of Galatzer-Levy and colleagues' [7] review was that the prevalence of participants in the *resilience* trajectory was significantly higher in *prospective* studies, defined as studies including data from prior to trauma exposure, relative to *longitudinal* studies, defined as studies with posttrauma data only. The authors suggested that this finding indicates selection bias in cohorts recruited in the aftermath of trauma, such that those suffering from mental health symptoms are likely to be overrepresented. An alternative explanation would be the opposite, i.e., that those suffering from mental health symptoms are underrepresented in prospective studies. Notably, however, all of the studies defined as prospective included pre-trauma mental health symptom assessments in the trajectory analysis. Symptoms therefore were, logically, not assessed in reference to the traumatic event that was the focus of the study. The value of this approach is elucidating various pre- to posttrauma patterns of symptoms over time, yet it is at odds with a conceptualization of PTSS as tied to a specific event. It is also possible that the lower rates of a resilience trajectory observed in these studies were due to this method of symptom assessment, rather than–or in addition to–selection bias.

An alternative means of incorporating prospective data is to explore pretrauma factors as predictors of posttrauma symptom trajectories, including trajectories of PTSS tied to the focal traumatic event. Information on pretrauma factors that increase risk for trajectories characterized by short- or long-term symptom elevations could assist in identifying vulnerable individuals or groups in the immediate aftermath or even prior to the occurrence of a disaster. Predisaster resources and vulnerabilities could be assessed retrospectively after a disaster, but would be biased by survivors' disaster-related experiences and postdisaster symptoms. To our knowledge, no study to date has explored prospective predictors of PTSS trajectories, disaster-related or otherwise. The broader disaster mental health literature, however, suggests at least three predisaster factors as potentially important in shaping postdisaster PTSS trajectories: perceived social support, predisaster mental illness, and predisaster physical health.

First, a large body of research has linked higher postdisaster perceived social support to lower postdisaster PTSS [1, 2]. Pathways from perceived social support and mental health symptoms, including PTSS, are thought to be bidirectional. On the one hand, consistent with *social causation*, low social support could increase vulnerability to symptoms; on the other, consistent with *social selection*, symptoms could impede the development and maintenance of supportive social relationships or lead one to perceive their relationships as less supportive [8]. Both mechanisms have received empirical support in the aftermath of disasters, with patterns of results varying depending on the nature of support provided and timing of assessment [9, 10]. Postdisaster social networks, irrespective of their relationship with postdisaster mental health symptoms, are unlikely to consist entirely of newly developed relationships, however, but rather are likely to depend in large part on the strength of the survivor's predisaster social relationships. Although few studies have included data on predisaster social support, those that have provide mixed evidence on its role in shaping postdisaster mental health. For example, predisaster perceived social support was included in the Resilience in Survivors of Katrina (RISK) study, a prospective study of low-income, predominantly Black, mothers originally recruited as part of a community college intervention in New Orleans prior to Hurricane Katrina [11]. At the univariate level, higher predisaster perceived social support in the RISK study was significantly associated with lower PTSS at one year postdisaster, but not significantly associated with PTSS four years postdisaster [12]. Such inconsistency could reflect the greater role of predisaster support in shaping the likelihood of some postdisaster symptom trajectories versus others (e.g., those characterized by short-term versus longer-term symptom elevations). At least two prospective studies to date have found predisaster support to be associated with mental health symptom trajectory membership [13, 14]. However, both studies included both pre and postdisaster symptoms in estimating trajectories and, as such, neither study assessed disaster-related PTSS. Differences in predisaster support across the trajectories in these studies thus could have been driven by variability in predisaster mental health.

Second, prior research has provided robust evidence that predisaster mental illness is significantly predictive of postdisaster psychiatric symptoms [15–18]. Such associations have been documented across studies with a wide range of postdisaster time frames, ranging from one month to five years postdisaster, suggesting that predisaster mental illness might confer risk for trajectories marked by both temporarily and chronically elevated PTSS. In this vein, a study of older adults after Hurricane Ike found that having a history of psychopathology was associated with increased risk for both chronically high and delayed-onset postdisaster PTSS trajectories, relative to a consistently low PTSS trajectory [19]. However, history of psychopathology was assessed retrospectively at the first postdisaster assessment and could have been biased by concurrent symptoms.

Third, a more limited body of research has provided mixed evidence regarding whether predisaster physical health could serve as a vulnerability factor for trajectories that include

elevated PTSS. At least two studies have found non-significant associations between predisaster markers of physical health–including the presence of one or more chronic diseases (e.g., high blood pressure, allergies, arthosis) [20]; a count of previously diagnosed medical conditions (diabetes, asthma, hypertension and any other medical condition) [21]; and a self-rated assessment of general health [18]–and postdisaster mental health. Conversely, Momma and colleagues [22] found predisaster leg extension power–a marker of physical functioning–to be negatively associated with PTSS after the Great East Japan Earthquake. The aforementioned study of older adults after Hurricane Ike further found predisaster serious illness or injury, retrospectively assessed, to be associated with increased risk for chronically elevated and delayed-onset PTSS trajectories, relatively to a consistently low PTSS trajectory [19].

When considering the role of predisaster risk and protective factors in shaping the likelihood of varied postdisaster PTSS trajectories, it is important to take into account disaster exposure. The ways in which disaster exposure is measured varies considerably across studies, but various indicators, including counts of disaster-related trauma (e.g., limited access to life-sustaining resources, perceived life threat) and specific experiences (e.g., bereavement, property damage), have been linked to adverse postdisaster mental health outcomes [2, 23]. Although research on predisaster predictors of disaster exposure is lacking, prior analyses of the RISK study have shown that participants with lower predisaster support and higher predisaster psychological distress tended to face exposure to a greater number of disaster-related trauma [24, 25]. These associations, in turn, could account for their enhanced risk for PTSS. Indeed, RISK analyses have shown lower perceived social support to be indirectly associated with higher postdisaster PTSS via exposure to a greater number of disaster-related trauma (12). It is unclear how such relationships could influence the likelihood of various postdisaster symptom trajectories. One possibility is that disaster exposure only attenuates the influence of predisaster factors on trajectories characterized by short-term elevations in symptoms. Alternatively, disaster exposure could attenuate associations between predisaster factors on the likelihood of all non-resilient trajectories.

In addition to lacking information on predisaster predictors, a further limitation of the literature on postdisaster PTSS trajectories is that the timeframe of most studies is limited to the first few postdisaster years [23]. On the one hand, a focus on the short-term aftermath aligns with the often time-limited services offered to disaster survivors and thus could assist in maximizing their impact; on the other, it sheds little light on the proportion and characteristics of survivors who might be in need of ongoing assistance. Furthermore, lack of long-term follow-up limits knowledge regarding the timeframe during which predisaster factors are predictive of postdisaster mental health.

## The current study

The current study explored predisaster predictors of PTSS trajectories among RISK study participants. Participants were living in the New Orleans area at the time of Hurricane Katrina and had completed a baseline survey as part of a community college intervention study between November 2003 and February 2005. Hurricane Katrina then struck the Gulf Coast area on August 29, 2005, and led to 1,833 deaths and an estimated 125 billion US$ in damages [26]. Participants were re-interviewed three times after the hurricane, approximately 1 year, 4 years, and 12 years postdisaster. We first documented the proportion of participants with most likely membership in various postdisaster PTSS trajectories. Second, we assessed the role of predisaster resources and vulnerabilities in predicting the likelihood of membership in different PTSS trajectories. We ran predictive models first without and then with adjustment for indicators of disaster exposure to assess whether relationships between predisaster factors and

the odds of PTSS trajectories marked by symptoms elevations were attenuated in the latter as compared to the former. Such findings would suggest that the influence of predisaster factors on PTSS trajectory membership might be indirect via disaster exposure.

## Materials and methods

### Participants and procedures

RISK participants were originally recruited as part of the Opening Doors Study, an investigation of an educational intervention targeting low-income parents at community colleges throughout the United States [27]. To be eligible for the study, students at participating community colleges had to be a parent, between the ages of 18 and 34 years old, and earn less than 200% of the federal poverty line. Opening Doors participants were randomly assigned to either an intervention condition, in which they received extra advising and a $1,000 stipend for each subsequent semester, or a control condition, in which they received neither of these benefits [23]. The study included participants from two community colleges in the New Orleans area, and were predominantly low-income, non-Hispanic Black and unmarried mothers. Because both of the community colleges were closed during the Fall 2005 semester due to Hurricane Katrina, the New Orleans participants were dropped from the larger Opening Doors study. However, the researchers secured funding to follow them as part of the RISK study, focused on their postdisaster psychosocial wellbeing [11]. We note here that no differences in postdisaster mental health outcomes were observed between participants who were in the intervention and control conditions; as such, intervention status has been excluded from primarily analyses of the RISK data [18, 21, 28], and was excluded from the current study as well.

A total of 1,019 RISK participants (96.0% women, $n$ = 942) completed a survey of demographic characteristics and psychosocial resources and vulnerabilities upon enrollment in the Opening Doors study, between November 2003 and February 2005 (*baseline*). Participants were invited to complete telephone interviews at three postdisaster time points: between March 2006 and March 2007 (*Time 1; T1;* approximately one year postdisaster), between March 2009 and April 2010 (*Time 2; T2;* approximately four years postdisaster), and between November 2016 and December 2018 (*Time 3; T3;* approximately 12 years postdisaster). Of the 942 women in the baseline sample, 667 (70.8%) completed the T1 survey, 714 (75.8%) the T2 survey, and 715 (75.9%) the T3 survey. Each postdisaster survey included an assessment of PTSS. The current analysis was limited to female participants who completed at least one postdisaster assessment of PTSS and thus could be included in trajectory analysis; 885 of the 942 women in the baseline sample (93.7%) met this criterion. Participants provided written consent at baseline. Since interviews were conducted via telephone at T1-T3, oral consent was obtained and documented by interviewers. This approach to obtaining informed consent, as well as all other study procedures, were approved by the Institutional Review Board of Harvard University.

### Measures

**PTSS.** PTSS was assessed using the 22-item Impact of Events Scale-Revised (IES-R) [29], which has been shown to have good psychometric properties [30]. Participants indicated how often, over the past week, they were bothered by experiences related to the hurricane (e.g., "pictures about it popped into my mind," "I was jumpy and easily startled"). Response options included "not at all" (0), "a little bit" (1), "moderately (2), "quite a bit" (3), and "extremely" (4). In accordance with the scale instructions [29], scale scores were computed as a mean of all items. Cronbach's alpha ($\alpha$) of reliability in the current study was .94 at T1, .95 at T2, and .99 at T3.

**Predisaster resources and vulnerabilities.** Perceived social support, conceptualized as a predisaster resource, was assessed at baseline using an eight-item version of the Social Provisions Scale (SPS) [31]. Items (e.g., "I have a trustworthy person to turn to if I have problems," "There are people I know will help me if I really need it") were rated from 1 (*strongly disagree*) to 4 (*strongly agree*). Scale scores were computed as the mean of all items (α = .78).

Two predisaster vulnerabilities were assessed at baseline. First, participants completed the six-item Kessler scale (K6) [32], which measures non-specific psychological distress and has good psychometric properties [33]. Participants indicated about how often over the past month they felt each indicator of distress (e.g., "nervous," "worthless") from 0 (*none of the time*) to 4 (*all of the time*), and scale scores were computed as the sum of all items (α = .76). Scores of 8–12 are indicative of probable mild or moderate mental illness, and scores of 13 or greater are indicative of probable serious mental illness [34]. In the current study, a dummy code for whether the participant had a score of 8 or higher was included to indicate probable mental illness at baseline.

Second, participants provided information about their physical health status. They indicated whether they had ever been diagnosed with five physical health conditions (asthma, high cholesterol, high blood pressure or hypertension, a heart condition, diabetes), or any other medical condition. Additionally, participants indicated whether they were currently had four physical health problems (back problems, digestive problems, frequent headaches or migraines, and anemia). A total count of physical health conditions or problems was computed.

**Disaster exposure.** Participants reported on exposure to Hurricane Katrina at both T1 and T2. For participants who completed both assessments, their responses from T1 were utilized. Three indicators of exposure were included. First, participants completed an eight-item checklist of disaster-related trauma, with items drawn from a larger survey of Katrina evacuees [35]. The eight items were: (1) lacked enough fresh water to drink, (2) lacked enough food to eat, (3) felt one's life was in danger, (4) lacked necessary medicine, (5) lacked necessary medical care, (6) family member lacked necessary medical care, (7) lacked knowledge of safety of children, and (8) lacked knowledge about safety of other family members. A count of affirmative responses was included. Second, participants indicated (Yes/No) whether they had lost a family member or close friend due to the hurricane and its aftermath (*bereavement*). Lastly, participants reported on the extent to which their predisaster home was damaged, ranging from 0 (*none*) to 4 (*enormous*).

**Demographic characteristics.** Demographic characteristics assessed at baseline included the participant's age in years, race/ethnicity (1 = Non-Hispanic Black; 0 = Other), number of children, and relationships status (1 = married and/or cohabiting; 0 = unmarried and not cohabiting). We note here that, of the 855 participants in the analytic sample who reported on race/ethnicity at baseline, 85 (9.9%) identified as non-Hispanic white, 20 (2.3%) as Hispanic, and 14 (1.6%) as other race/ethnicity. Because of these small subsample sizes, especially of the latter two groups, these categories were combined into the larger "Other" classification. Additionally, participants reported on whether they received the following benefits: (1) unemployment or dislocated workers benefits, (2) supplemental security income or disability, (3) cash assistance or welfare (TANF), and (4) food stamps. A count of benefits received was computed. These were included based on prior research showing links to postdisaster mental health [1, 2].

## Statistical analysis

Data analysis consisted of three steps. First, a series of preliminary analyses were conducted. Descriptive data were computed for the 885 women in the analytic sample. Independent-

samples *t*-tests and chi-square analyses assessed for differences between the 885 participants in the analytic sample and the 57 women from the baseline sample who were dropped due to missing data.

Second, latent class growth analysis (LCGA) was conducted. Models were run with T1 anchored at 1, T2 at 4, and T3 at 12, representing the approximate number of postdisaster years, and included both linear and quadratic growth terms. We examined solutions with 1 to 10 classes, reviewing both statistical criteria (Akaike Information Criterion [AIC], Bayesian Information Criterion [BIC], and sample size adjusted BIC), with lower values indicating better fit; entropy and mean posterior probabilities, with higher values indicating greater classification quality; and the Lo-Mendel-Rubin likelihood ratio test [LMR-LRT], which compares whether a model with $k + 1$ significantly improves fit over a model with $k$ classes) and theoretical criteria (e.g., interpretability, parsimony, and clinical significance) as per recommended guidelines [36–38]. We also recorded the number and percentage of participants with most likely membership in each class for each solution. Once the model that best represented the data was selected, descriptive data for the subsamples of participants with most likely membership in each trajectory were computed.

Third, the three-step method was used to assess relationships between predictors and trajectory membership [39, 40]. This procedure is similar to multinomial logistic regression, but accounts for the uncertainty in trajectory membership. Univariate analyses were first run for each predictor. Subsequently, two multivariable models were conducted. Model 1included only baseline variables (demographic characteristics: age, non-Hispanic Black race, number of children, married or cohabiting, and number of benefits; and predisaster resources and vulnerabilities: perceived social support, number of physical health conditions or problems, and probable moderate or severe mental illness). Model 2 included the same variables as Model 1, as well as indicators of disaster exposure (number of hurricane-related traumatic events, bereavement, and housing damage).

Missing data on PTSS was estimated via full information maximum likelihood in the LCGA models. Missing data on predictors was handled using multiple imputation. Ten datasets with complete information on predictors for the 885 participants were computed, and results for the pooled analysis with correction for standard error are presented. For the LCGA and predictive models, estimates, standard errors, and 95% confidence intervals (CIs) were estimated using 1,000 bootstrapped samples. Data management and attrition analyses were conducted in SPSS 25.0 [41], and all other analyses were conducted in Mplus 8.0 [42].

## Results

### Preliminary analyses

Table 1 shows descriptive statistics for the 885 women in the analytic sample. Women were on average 25.19 years (*SD* = 4.45) and had 1.81 children (*SD* = 1.03) at baseline. The majority (86.2%) identified as non-Hispanic Black, and 22.9% were either married or cohabiting at baseline. On average, participants reported 3.01 (*SD* = 2.29) disaster-related traumas and 30.1% experienced bereavement. Participants' average PTSS scores decreased over the course of the study.

Participants in the analytic sample had significantly higher baseline social support than those who were excluded due to missing data, $t(976) = 2.52$, $p = .012$. All other differences did not reach statistical significance.

### Latent class growth analysis

Table 2 shows the results of the LCGA models. Based on recommended criteria, the model with three classes was selected as the best representation of the data. Several factors went into

**Table 1. Descriptive data for participants in the analytic sample.**

|  |  | *M (SD) or %* |
|---|---|---|
| Demographic characteristics |  |  |
|  | Age | 25.19 (4.45) |
|  | Non-Hispanic Black | 86.2% |
|  | Number of children | 1.81 (1.03) |
|  | Married or cohabiting | 22.9% |
|  | Number of benefits | 0.92 (0.71) |
| Predisaster resources and vulnerabilities |  |  |
|  | Perceived social support | 3.20 (0.45) |
|  | Probable mental illness | 23.1% |
|  | Number of physical health conditions or problems | 0.86 (1.05) |
| Disaster exposure |  |  |
|  | Number of disaster-related trauma | 3.01 (2.29) |
|  | Bereavement | 30.1% |
|  | Housing damage (0–4) | 2.88 (1.15) |
| PTSS |  |  |
|  | Time 1 | 1.51 (0.99) |
|  | Time 2 | 1.24 (0.96) |
|  | Time 3 | 0.72 (0.90) |

*Note*. PTSS = Posttraumatic Stress Symptoms. PTSS scores were computed using raw data from subsamples of participants who provided data at each wave (Time 1 $n$ = 667; Time 2 $n$ = 714; Time 3 $n$ = 711). All other values reflect the combined results from 10 multiple imputations ($N$ = 885).

this selection. The models with five or more classes were eliminated from consideration due to the small number and percentage of participants in the smallest class, ranging from 9 to 11 participants (1.0–1.2% of the sample). For the remaining models, we noted that the AIC, BIC, and adjusted BIC consistently decreased, but leveled off after the two-class solution; that entropy was highest for the four-class solution and average posterior probabilities for the two-class solution; and that the significance level of the LMR-LRT was $p < .05$ for both the two- and three-class solutions. We then inspected plots of estimated means for the two-, three-, and four-class solutions. We noted that the three-class model had trajectories that were both distinctive, consistent with prior theory and research, and clinically meaningful. In contrast, the two least common trajectories in four-class solution were very similar to each other in shape (both were characterized by initially high symptoms that did not significant change over time), and the differences between them were determined to lack clinical significance. The three-class solution was therefore selected as the best representation of the data. Additional information on model selection, including plots of estimated means for each solution, is available upon request.

Table 3 provides descriptive data for growth terms and estimated means at each wave for the three trajectories. Fig 1 shows a plot of estimated means at each wave for each trajectory from the three-class model. We note here that we have included dashed or dotted lines to connect data points over time for each trajectory to illustrate changes in PTSS across the three postdisaster waves. It is worth emphasizing, however, that this depiction does not capture the likely non-linear changes in PTSS between waves, which would require additional data points over the study period.

Trajectories were named based on descriptive results, specifically the correspondence between mean PTSS scores for participants with most likely membership in the given

**Table 2. Results of latent class growth models.**

| N classes | AIC | BIC | Adj. BIC | Entropy | M Posterior probability (SD) | p of LMR-LRT |
|---|---|---|---|---|---|---|
| 1 | 5714.40 | 5743.11 | 5724.05 | – | – | N/A |
| 2 | 5287.19 | 5335.05 | 5303.29 | .801 | .93 (.04) | < .001 |
| **3** | **5112.99** | **5179.95** | **5135.49** | **.813** | **.90 (.05)** | **.018** |
| 4 | 4995.49 | 5081.63 | 5024.46 | .816 | .88 (.03) | .126 |
| 5 | 4915.06 | 5020.34 | 4950.47 | .837 | .91 (.07) | .049 |
| 6 | 4808.35 | 4932.77 | 4850.20 | .836 | .91 (.07) | .011 |
| 7 | 4760.38 | 4903.95 | 4808.76 | .826 | .89 (.07) | .263 |
| 8 | 4720.50 | 4883.21 | 4775.23 | .832 | .90 (.07) | .110 |
| 9 | 4661.36 | 4843.21 | 4722.53 | .826 | .90 (.07) | .094 |
| 10 | 4630.63 | 4831.63 | 4698.24 | .828 | .89 (.04) | |

| N classes | Number and Percentage of Participants in Each Class | | | | | | | |
|---|---|---|---|---|---|---|---|---|
| | 1 | 2 | 3 | 4 | 5 | 6 | 7 | 8 |
| 1 | 885 (100.0%) | – | – | – | – | – | – | – |
| 2 | 204 (23.1%) | 681 (76.8%) | – | – | – | – | – | – |
| **3** | **67 (7.6%)** | **204 (23.1%)** | **614 (69.4%)** | – | – | – | – | – |
| 4 | 47 (5.3%) | 100 (11.3%) | 162 (18.3%) | 576 (65.1%) | – | – | – | – |
| 5 | 9 (1.0%) | 49 (5.5%) | 122 (13.8%) | 138 (15.6%) | 567 (64.1%) | – | – | – |
| 6 | 9 (1.0%) | 44 (5.0%) | 62 (7.0%) | 110 (12.4%) | 112 (12.7%) | 548 (61.9%) | – | – |
| 7 | 9 (1.0%) | 44 (5.0%) | 54 (6.1%) | 74 (8.4%) | 75 (8.5%) | 89 (10.1%) | 540 (61.0%) | – |
| 8 | 9 (1.0%) | 24 (2.7%) | 27 (3.1%) | 46 (5.2%) | 75 (8.5%) | 78 (8.8%) | 90 (10.2%) | 536 (60.6%) |
| 9 | 9 (1.0%) | 25 (2.8%) | 27 (3.1%) | 46 (5.2%) | 61 (6.9%) | 65 (7.3%) | 67 (7.6%) | 72 (8.1%) |
| 10 | 11 (1.2%) | 22 (2.5%) | 23 (2.6%) | 38 (4.3%) | 44 (5.0%) | 46 (5.2%) | 59 (6.7%) | 62 (7.0%) |

| N classes | Number and Percentage of Participants in Each Class | |
|---|---|---|
| | 9 | 10 |
| 1 | – | – |
| 2 | – | – |
| **3** | – | – |
| 4 | – | – |
| 5 | – | – |
| 6 | – | – |
| 7 | – | – |
| 8 | – | – |
| 9 | 513 (58.0%) | – |
| 10 | 67 (7.6%) | 513 (58.0%) |

*Note.* N = 885. Adj. = Adjusted; AIC = Akaike Information Criterion; BIC = Bayesian Information Criterion; LMR-LRT = Lo-Mendel-Rubin Likelihood Ratio Test.

trajectory at each time point and IES-R item ratings, and the statistical significance of growth terms. The majority of participants had their most likely membership in a *Moderate-Decreasing* trajectory (69.3%, n = 614). The *Moderate-Decreasing* trajectory was characterized by average symptom ratings between *a little* (1) and *moderately* (2) at T1, and between *not at all* (0) and *a little* (1) at T2 and T3, and a significantly negative linear slope (*Est.* = -0.10, *SE* = 0.03, 95% confidence interval [CI]: -0.15 –-0.05, *p* < .001). Nearly a quarter had their most likely membership in a *High-Decreasing* trajectory (23.1%, n = 204). Participants in the *High-Decreasing* trajectory had average symptom ratings between *moderately* (2) and *quite a bit* (3) at T1, and *a little* (1) and *moderately* (2) at T2 and T3, and significant negative linear growth (*Est* = -0.12, *SE* = 0.05, 95% CI: -0.12 –-0.03, *p* < .001). Less than ten percent had their most likely membership in a *High-Stable* trajectory (7.6%, n = 67). On average, participants in the

**Table 3. Descriptive statistics for growth terms and estimated posttraumatic stress symptoms (PTSS) for three-class model.**

| | | *Moderate-Decreasing (69.3%, n = 614)* | *High-Decreasing (23.1%, n = 204)* | *High-Stable (7.6%, n = 67)* |
|---|---|---|---|---|
| | | *M (SE), 95% CI* | *M (SE), 95% CI* | *M (SE), 95% CI* |
| Growth Terms | | | | |
| | Intercept | 1.31 (0.06) [1.19, 1.43]*** | 2.15 (0.12) [1.91, 2.39]*** | 2.04 (0.24) [1.57, 2.50]*** |
| | Slope | -0.10 (0.03) [-0.15, -0.05]*** | -0.12 (0.05) [-0.21, -0.03]*** | 0.05 (0.08) [-0.10, 0.21] |
| | Quadratic | < 0.01 (< 0.01) [>-0.01, <0.01] | 0.01 (<0.01) [>-0.01, 0.01] | < 0.01 (0.01) [-0.01, 0.01] |
| PTSS | | | | |
| | Time 1 | 1.21 (0.05) [1.12, 1.30] | 2.04 (0.10) [1.84, 2.24] | 2.09 (0.20) [1.71, 2.47] |
| | Time 2 | 0.92 (0.04) [0.84, 1.01] | 1.75 (0.11) [1.54, 1.96] | 2.26 (0.17) [1.92, 2.59] |
| | Time 3 | 0.18 (0.02) [0.14, 0..22] | 1.45 (0.10) [1.26, 1.64] | 2.77 (0.16) [2.46, 3.08] |

*Note.* N = 885.

*** $p < .001$.

*High-Stable* trajectory rated symptoms between *moderately* (2) and *quite a bit* (3) at all three postdisaster time points. Neither the linear nor quadratic growth term reached statistical significance for the *High-Stable* trajectory (*Est.* = 0.05, *SE* = 0.08, 95% CI: -0.10–0.21, *p* = .496, and *Est.* < 0.01, *SE* = 0.01, 95% CI: -0.01–0.01, *p* = .887, respectively). Table 4 shows

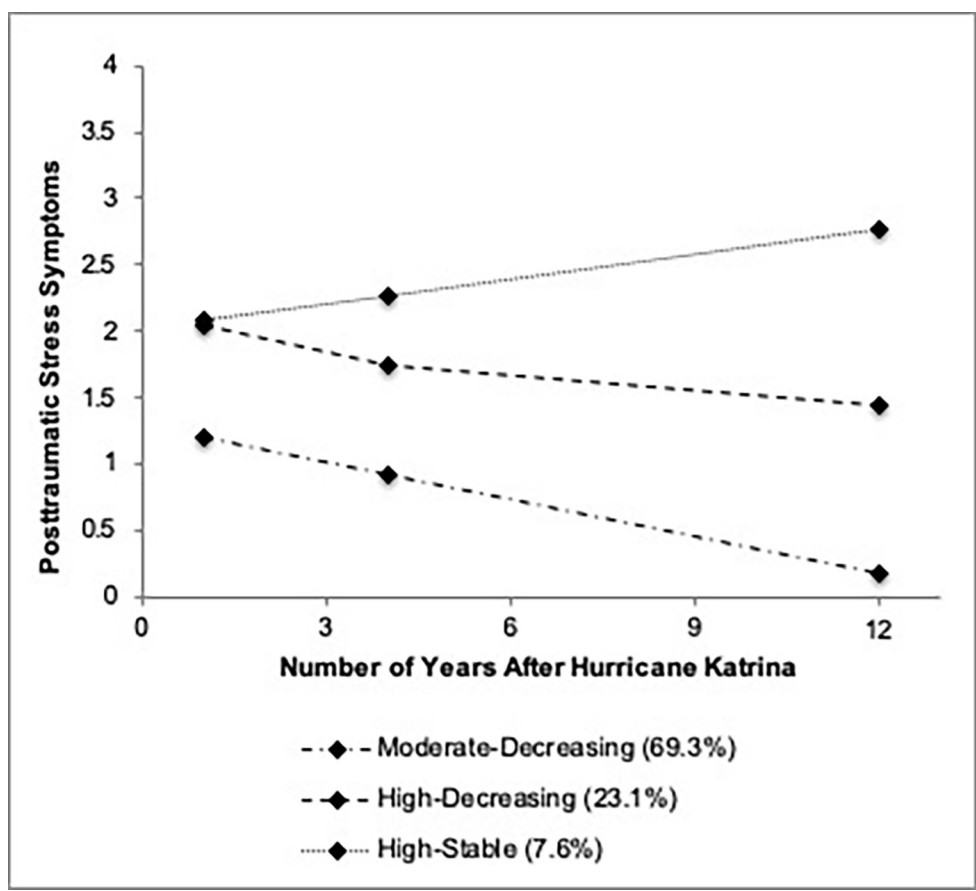

**Fig 1. Plot of estimated means for posttraumatic stress symptom (PTSS) trajectories from the three-class model.**

**Table 4. Descriptive statistics for participants with most likely membership in each posttraumatic stress symptom trajectory.**

| | Moderate-Decreasing (69.3%, n = 614) | High-Decreasing (23.1%, n = 204) | High-Stable (7.6%, n = 67) |
|---|---|---|---|
| | M (SD) or % | M (SD) or % | M (SD) or % |
| Demographic characteristics | | | |
| Age | 24.93 (4.29) | 25.47 (4.90) | 26.75 (4.12) |
| Non-Hispanic Black | 83.5% | 91.1% | 95.5% |
| Number of children | 1.78 (1.01) | 1.78 (0.96) | 2.13 (1.45) |
| Married or cohabiting | 23.8% | 21.3% | 19.6% |
| Number of benefits | 0.89 (0.71) | 0.96 (0.72) | 1.15 (0.61) |
| Predisaster resources and vulnerabilities | | | |
| Perceived social support | 3.23 (0.45) | 3.12 (0.43) | 3.16 (0.44) |
| Number of physical health conditions or problems | 0.80 (1.01) | 0.90 (1.10) | 1.26 (1.19) |
| Probable moderate or severe mental illness | 19.4% | 30.5% | 37.9% |
| Hurricane exposures | | | |
| Number of hurricane-related traumatic events | 2.63 (2.13) | 3.77 (2.39) | 4.21 (2.40) |
| Bereavement | 24.6% | 40.4% | 49.7% |
| Housing damage (0–4) | 2.76 (1.15) | 3.17 (1.09) | 3.10 (1.26) |

*Note.* N = 885.

descriptive statistics for all predictor variables for the three subsamples with most likely membership in each trajectory.

## Predictive models

Table 5 shows the results of univariate analyses and multivariable models predicting trajectory membership, run using the three-step procedure. In the univariate analyses, all three predisaster factors showed significant relationships with trajectory membership. Predisaster perceived social support was significantly associated with lower odds of the *High-Decreasing* trajectory, versus the *Moderate-Decreasing* trajectory, such that each unit increase on the perceived social support scale was associated with 0.52 (95% CI: 0.33–0.82) the odds of membership in the *High-Decreasing* trajectory, relative to the *Moderate-Decreasing* trajectory. Predisaster physical health conditions or problems were associated with increased odds of the *High-Stable* trajectory, versus both the *High-Decreasing* and *Moderate-Decreasing* trajectory. Specifically, each additional physical health condition or problem was associated with 1.49 (95% CI: 1.19–1.85) and 1.37 (95% CI: 1.02–1.84) the odds of the *High-Stable* trajectory, relative to the *Moderate-Decreasing* and *High-Decreasing* trajectories, respectively. Lastly, probable predisaster mental illness was significantly associated with increased odds of both the *High-Decreasing* and *High-Stable* trajectories, versus the *Moderate-Decreasing* trajectory, such that participants with probable predisaster mental illness had 1.97 the odds (95% CI: 1.23–3.17) of the *High-Decreasing* trajectory and 2.75 the odds (95% CI: 1.49–5.07) of the *High-Stable* trajectory, relative to the *Moderate-Decreasing* trajectory.

In Model 1, without adjustment for disaster exposure, probable predisaster mental illness was associated with increased odds of a *High-Decreasing* trajectory, versus a *Moderate-Decreasing* trajectory. Specifically, participants with predisaster probable mental illness had 1.75 odds (95% CI: 1.03–2.97) of membership in the *High-Decreasing* trajectory, relative to the *Moderate-Decreasing* trajectory. In addition, probable predisaster mental illness and more predisaster physical health conditions or problems were associated with increased odds of the *High-Stable* trajectory, relative to the *Moderate-Decreasing* trajectory. Probable predisaster mental illness

**Table 5. Results of univariate analyses and multivariable models predicting trajectory membership.**

| | High-Decreasing (n = 204) vs. Moderate-Decreasing (n = 614) | | |
|---|---|---|---|
| | Univariate Analysis | Model 1 | Model 2 |
| Demographic characteristics | | | |
| Age | 1.03 (0.98–1.08) | 1.05 (0.99–1.11) | 1.04 (0.98–1.11) |
| Non-Hispanic Black | 2.22 (1.17–4.47)* | 2.33 (1.10–4.93)* | 1.60 (0.71–3.62) |
| Number of children | 0.98 (0.80–1.20) | 0.88 (0.71–1.10) | 0.84 (0.67–1.05) |
| Married or cohabiting | 0.85 (0.52–1.38) | 0.91 (0.54–1.52) | 1.08 (0.62–1.88) |
| Number of benefits | 1.17 (0.87–1.56) | 1.07 (0.78–1.45) | 0.96 (0.68–1.36) |
| Predisaster resources and vulnerabilities | | | |
| Perceived social support | 0.52 (0.33–0.82)** | 0.61 (0.38–1.00) | 0.75 (0.43–1.28) |
| Number of physical health conditions or problems | 1.08 (0.89–1.33) | 1.03 (0.83–1.28) | 0.95 (0.75–1.21) |
| Probable moderate or severe mental illness | 1.97 (1.23–3.17)** | 1.75 (1.03–2.97)* | 1.79 (0.97–3.28) |
| Hurricane exposures | | | |
| Number of hurricane-related traumatic events | 1.30 (1.17–1.44)*** | – | 1.21 (1.08–1.36)** |
| Bereavement | 2.31 (1.45–3.68)*** | – | 1.78 (1.05–3.02)* |
| Housing damage (0–4) | 1.49 (1.20–1.84)*** | – | 1.35 (1.06–1.70)* |
| | High-Stable (n = 67) vs. Moderate-Decreasing (n = 614) | | |
| Demographic characteristics | | | |
| Age | 1.11 (1.04–1.18)** | 1.11 (1.03–1.20)** | 1.10 (1.02–1.18)* |
| Non-Hispanic Black | 4.83 (1.18–19.85)* | 4.72 (0.86–26.03) | 3.58 (0.43–29.84) |
| Number of children | 1.36 (1.09–1.69)** | 1.15 (0.89–1.49) | 1.08 (0.79–1.46) |
| Married or cohabiting | 0.76 (0.38–1.54) | 0.77 (0.36–1.66) | 0.92 (0.42–2.01) |
| Number of benefits | 1.72 (1.23–2.39)** | 1.54 (1.05–2.26)* | 1.45 (0.95–2.23) |
| Predisaster resources and vulnerabilities | | | |
| Perceived social support | 0.71 (0.38–1.32) | 1.12 (0.56–2.25) | 1.41 (0.65–3.07) |
| Number of physical health conditions or problems | 1.49 (1.19–1.85)*** | 1.41 (1.08–1.83)* | 1.30 (0.99–1.71) |
| Probable moderate or severe mental illness | 2.75 (1.49–5.07)*** | 2.92 (1.40–6.11)** | 3.03 (1.37–6.66)** |
| Hurricane exposures | | | |
| Number of hurricane-related traumatic events | 1.39 (1.21–1.60)*** | – | 1.26 (1.08–1.48)** |
| Bereavement | 3.35 (1.81–6.19)*** | – | 2.23 (1.13–4.41)* |
| Housing damage (0–4) | 1.31 (0.96–1.80) | – | 1.07 (0.74–1.54) |
| | High-Stable (n = 67) vs. High-Decreasing (n = 204) | | |
| Demographic characteristics | | | |
| Age | 1.08 (1.00–1.17) | 1.06 (0.97–1.16) | 1.05 (0.96–1.15) |
| Non-Hispanic Black | 2.17 (0.43–11.03) | 2.03 (0.28–14.70) | 2.24 (0.19–26.75) |
| Number of children | 1.39 (1.06–1.82)* | 1.30 (0.95–1.77) | 1.29 (0.93–1.79) |
| Married or cohabiting | 0.90 (0.38–2.10) | 0.85 (0.33–2.16) | 0.85 (0.33–2.19) |
| Number of benefits | 1.47 (0.98–2.21) | 1.44 (0.89–2.36) | 1.51 (0.88–2.57) |
| Predisaster resources and vulnerabilities | | | |
| Perceived social support | 1.37 (0.65–2.87) | 1.82 (0.80–4.17) | 1.90 (0.76–4.73) |
| Number of physical health conditions or problems | 1.37 (1.02–1.84)* | 1.36 (0.98–1.89) | 1.37 (0.97–1.93) |
| Probable moderate or severe mental illness | 1.40 (0.68–2.88) | 1.67 (0.71–3.93) | 1.70 (0.68–4.23) |
| Hurricane exposures | | | |
| Number of hurricane-related traumatic events | 1.07 (0.92–1.25) | – | 1.04 (0.86–1.26) |
| Bereavement | 1.45 (0.68–3.08) | – | 1.25 (0.54–2.87) |

(*Continued*)

**Table 5.** (Continued)

| | | | | |
|---|---|---|---|---|
| | Housing damage (0–4) | 0.88 (0.60–1.30) | – | 0.79 (0.51–1.23) |

*Note*. N = 885.

\* *p* < .05

\*\* *p* < .01

\*\*\* *p* < .001.

was associated with 2.92 odds (95% CI: 1.40–6.11) and each additional predisaster physical health condition or problems with 1.41 odds (95% CI: 1.08–1.83) of membership in the *High-Stable* trajectory, relative to the *Moderate-Decreasing* trajectory.

After adjustment for hurricane exposures in Model 2, none of the predisaster factors was significantly associated with the odds of the *High-Decreasing* trajectory, versus the *Moderate-Decreasing* trajectory. In contrast, each disaster exposure was associated with increased odds of the *High-Decreasing* trajectory, versus the *Moderate-Decreasing* trajectory, such that each additional disaster-related trauma was associated with 1.21 odds (95% CI: 1.08–1.36), bereavement with 1.78 odds (95% CI: 1.05–3.02), and each additional unit on the housing damage rating scale with 1.35 odds (95% CI: 1.06–1.70) of the *High-Decreasing* trajectory, versus the *Moderate-Decreasing* trajectory. Probable predisaster mental illness remained significantly associated with the odds of the *High-Stable* trajectory, relative to the *Moderate-Decreasing* trajectory, such that being classified as having a probable predisaster mental illness was associated with 3.03 odds (95% CI: 1.37–6.66) of the *High-Stable* trajectory, versus the *Moderate-Decreasing* trajectory. Disaster-related trauma and bereavement were significant predictors of the *High-Stable* trajectory, relative to the *Moderate-Decreasing* trajectory. Each additional disaster-related trauma was associated with 1.26 odds (95% CI: 1.08–1.48), and bereavement with 2.23 odds (95% CI: 1.13–4.42) of the *High-Stable* trajectory, relative to the *Moderate-Decreasing* trajectory.

## Discussion

The current study is the first to our knowledge to examine pretrauma predictors of PTSS trajectories using prospective data. Specifically, we explored predisaster perceived social support, probable mental illness, and physical health conditions or problems as predictors of PTSS trajectories among a sample of low-income, primarily non-Hispanic Black, mothers who experienced Hurricane Katrina. Participants were surveyed prior to Hurricane Katrina, between November 2003 and February 2005, as part of their involvement in a community college intervention study, and were re-interviewed 1 year, 4 years, and 12 years after the hurricane made landfall in the New Orleans area on August 29, 2005. Three PTSS trajectories were detected, with the majority of participants (69.3%) in a trajectory characterized by moderate symptoms that decreased over time (*Moderate-Decreasing*), nearly a quarter (23.1%) in a trajectory of high symptoms that decreased over time (*High-Decreasing*), and less than ten percent (7.6%) in a trajectory of consistently high symptoms (*High-Stable*). Whereas predisaster perceived social support was not significantly associated with PTSS trajectory membership in multivariable models, participants with probable predisaster mental illness had significantly greater odds of being in both the *High-Decreasing* and *High-Stable* trajectories, and those with a greater number of predisaster physical health conditions or problems had significantly greater odds of being in the *High-Stable* trajectory, relative to the *Moderate-Decreasing* trajectory. However, in a model that adjusted for disaster exposure, only probable predisaster mental

illness remained significantly associated with increased odds of being in the *High-Stable* trajectory, relative to the *Moderate-Decreasing* trajectory. The results suggest that, whereas predisaster mental illness confers risk for chronic PTSS in the aftermath of disaster, the impacts of the other predisaster resources and vulnerabilities included in this study on elevated PTSS trajectories do not go beyond those of disaster exposure.

The results regarding predisaster mental illness are consistent with prior research linking preexisting psychiatric symptoms with higher PTSS at various postdisaster time points [15–17]. The added contribution of the current study is evidence that those with preexisting probable mental illness were at risk for the *High-Stable* trajectory independent of disaster exposure, which was not the case for the *High-Decreasing* trajectory. This pattern of results suggests that the effect of predisaster mental illness on the risk for a trajectory characterized by short-term elevations in PTSS is mediated by disaster exposure, an interpretation that could be tested directly in future research. In contrast, the risk for chronic PTSS associated with predisaster mental illness, independent of disaster exposure, could indicate more longstanding difficulties that contribute to mental health problems, such as maladaptive coping strategies, or cumulative exposure to additional traumatic and stressful life events over the life-course. Again, these interpretations could be examined in future research.

The findings also add to the more limited research on the role of predisaster perceived social support in shaping posttrauma mental health trajectories. At the univariate level, predisaster perceived social support was associated with increased odds of the *High-Decreasing* trajectory, relative to the *Moderate-Decreasing* trajectory, consistent with prior findings in RISK showing relations with PTSS only at the earliest postdisaster wave [12]. However, associations between predisaster perceived social support and trajectory membership reduced to non-significance once demographic characteristics, predisaster probable mental illness, and predisaster physical health problems or conditions were adjusted. As such, it is possible that the role of predisaster perceived social support in shaping risk for short-term elevations in PTSS is via other predisaster factors, particularly predisaster mental health problems–since predisaster probable mental illness remained significantly associated with the *High-Decreasing* trajectory, versus the *Moderate-Decreasing* trajectory, in the model. A pathway from predisaster perceived social support to postdisaster PTSS via predisaster mental illness would be consistent with social causation. On the other hand, it is also possible that our pattern of results is indicative of social selection–if, for example, predisaster perceived social support and PTSS trajectories were only related in univariate models because they shared predisaster mental illness as a common predictor. Future prospective studies with multiple waves of predisaster data could shed additional light on this issue.

Unlike predisaster perceived social support, predisaster physical health conditions or problems remained a statistically significant predictor of trajectory membership–in this case, of the *High-Stable* PTSS trajectory, relative to the *Moderate-Decreasing* trajectory–when demographic characteristics and other predisaster factors were included in a multivariable model. However, this association was reduced to non-significance when indicators of disaster exposure were added. This pattern of results suggests that predisaster physical health conditions or problems influence postdisaster mental health by increasing risk for exposure, which could perhaps also explain the non-significant associations between predisaster physical health and postdisaster mental health in prior research [17, 18]. This indirect pathway could be explicitly tested in future research, and in doing so, researchers could parse out whether there are certain types of disaster-related exposures that are more common among those with preexisting physical health conditions or problems. For example, it is likely that those with preexisting physical health concerns are inherently more likely to face two of the disaster-related trauma on the inventory used in the current study–lack of necessary medicine, and lack of necessary medical

care. In a similar vein, future research could separately assess the influence of specific physical health conditions or problems, as well as interactions between various predisaster factors (e.g., between predisaster physical and mental health), on the likelihood of membership in different PTSS trajectories.

Although the focus of this study was predisaster predictors of PTSS trajectories, other findings are noteworthy. First, unlike most prior posttrauma trajectory studies [7], we did not find a trajectory of consistently low PTSS. The lack of what some might term a "resilient" trajectory could be a function of the sample characteristics–specifically, that all participants were female, parents, and low-income at baseline, and most were Black, characteristics that prior research has shown to increase risk for postdisaster psychiatric adversity [2]. This speaks to the value of conducting posttrauma trajectory studies within at-risk groups for greater insight into the circumstances under which resilience, defined as a trajectory of consistently low symptoms, is and is not the modal response to trauma. Notably, there have been other prior studies that also did not find a consistently low PTSS trajectory, and like our study, had modal trajectories defined by recovery from initially moderate symptoms. These studies included sexual assault survivors [43] and Palestinians exposed to chronic political violence [44], further indicating that what is commonly referred to as resilience might be limited in certain subpopulations and/or those facing especially severe trauma.

An additional explanation for the lack of a consistently low PTSS trajectory is that, because we followed an existing cohort, we were able to capture the psychological responses of disaster survivors who otherwise might unlikely to participate postdisaster research studies, such as those who were displaced or coping with major postdisaster stressors. In essence, our results could indicate selection bias in prior trajectory studies, but opposite that posited by Galatzer-Levy and colleagues' review [7]. It is also possible that the review's finding that resilience was less common in prospective, versus longitudinal, studies could have been due to the fact that mental health symptoms included in the former were not event-specific. Analyzing pre-to-posttrauma trajectories of mental health symptoms not tied to an event (e.g., major depression, non-specific psychological distress) alongside posttrauma trajectories of PTSS in the same prospective cohort would provide greater insight into this issue.

Second, in our final model, older age was associated with increased odds of the *High-Stable* trajectory, relative to the *Moderate-Decreasing* trajectory. The limited age range in the sample (18 to 34 years old at baseline) should be noted. It could be that the older women in the sample might have had other characteristics (e.g., lower socioeconomic status, lower parental support) that increased their risk for long-term adversity, or that disasters are generally more destabilizing for those who are further into early adulthood. Further research within this age group is warranted to shed light on this finding. Third, the three indicators of disaster exposure were predictive of trajectories of both temporarily and chronically elevated PTSS, relative to the *Moderate-Decreasing* trajectory. One exception was that housing damage was not associated with the *High-Stable* trajectory in both univariate and multivariable models, providing evidence that this type of exposure is only associated with short-term postdisaster symptoms. Lastly, none of the predictors in multivariable models were associated with increased odds of the *High-Stable* trajectory, relative to the *High-Decreasing* trajectory. This could be due to limited statistical power, as those two trajectories together comprised less than a third of the sample. It is likely, however, that intervening factors, such as subsequent trauma and stressor exposure–which were not included in the analysis–distinguish between disaster survivors who do and do not recover, which could be explored in future research. Future studies could also include changes in general psychological distress, physical health and perceived social support, as well as in demographic characteristics, including participants' access to social benefits,

marital and cohabitation status, and number of children, which could also potentially shape the odds of recovery from initially elevated postdisaster symptoms.

The results of this study have implications for disaster preparedness and response. First, they suggest the need to support those with preexisting mental health problems in the aftermath of disasters to prevent them from suffering from trajectories of chronic PTSS. Clinicians and others working with persons with mental illness should be sure to have up-to-date contact information for their patients, including mobile phone numbers and email addresses, so that they can easily connect with them in the aftermath of disaster. Efforts to promptly reconnect with patients could prevent treatment disruptions, for example by temporarily conducting sessions via telemedicine or web-based services, or by facilitating referrals for displaced or relocated survivors in their new communities. The results further suggest that clinicians working with disaster survivors suffering from PTSS, particularly in its chronic form, should attend to preexisting mental health problems. In doing so, clinicians should consider the influence of other risk factors for pre- and postdisaster mental health problems, including exposure to additional traumatic events and stressors experienced over the life-course, consistent with the recommendations for trauma-informed care [45].

The pattern of results further suggests the importance of reducing disaster exposure among those with preexisting mental and physical health problems. Such efforts could include making sure that patients–and potentially their close friends and family members as well–have plans in place for timely evacuation and means for staying connected should separations occur. Encouraging patients to have a backup supply of necessary medications and information on where they can access medical care in the event of disaster could also prevent potentially life-threatening exposures related to preexisting conditions. Those providing postdisaster assistance should similarly be sure to have adequate supplies of medication for common mental and physical health conditions, in addition to other life-sustaining resources.

This study has at least six limitations. First, PTSS was assessed via a self-report inventory, rather than a clinical assessment. This approach was efficient and consistent with several prior large-scale disaster studies [46, 47], however. Second, although recommended criteria were used to determine the trajectory model that best represented the data, the model selection process remains somewhat subjective. Third, we included a limited set of predisaster factors in the analysis, and future work should include other predisaster vulnerabilities and resources that could shape trajectory membership. For example, research has shown that prior exposure to traumatic and stressful life events is associated with adverse postdisaster mental health outcomes [48, 49]. Fourth, the time lag between the different study waves varied considerably, and eight years had passed between the second and third postdisaster assessments. The gaps between assessments mean that we do not have a complete picture of PTSS trajectories in the sample over the study period. It is possible that we would have identified a different set of PTSS trajectories, with distinctive patterns and predictors, if we had access to additional waves of data within the study timeframe. Fifth, although the low-income, predominantly Black, mothers in our sample were of interest given their risk for postdisaster adversity [2], they are not representative of Katrina survivors from New Orleans or disaster survivors more generally. They also are not representative of all low-income mothers living in New Orleans at the time of Katrina, given that they were enrolled in a community college intervention at baseline. Lastly, participants dropped from the analysis due to missing data had significantly lower baseline support than those who were included. This could indicate limited variance in predisaster support in the analysis, potentially attenuating associations with PTSS trajectory membership.

Despite these limitations, this study is the first to our knowledge to examine predisaster predictors of PTSS trajectories in the aftermath of disaster without reliance on retrospective assessment. The results demonstrate the vulnerability of persons suffering from predisaster

mental illness to chronic PTSS, and support outreach efforts targeting this group both leading up to and in the aftermath of disaster.

## Acknowledgments

The authors thank all of the individuals who participated in this project. They also thank Mariana Arcaya, Morgan Buchanan, and Meghan Zacher for their helpful feedback on this work.

## Author Contributions

**Conceptualization:** Sarah R. Lowe.

**Formal analysis:** Sarah R. Lowe, Ethan J. Raker.

**Funding acquisition:** Mary C. Waters, Jean E. Rhodes.

**Methodology:** Sarah R. Lowe, Mary C. Waters, Jean E. Rhodes.

**Writing – original draft:** Sarah R. Lowe, Ethan J. Raker.

**Writing – review & editing:** Mary C. Waters, Jean E. Rhodes.

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
