## [Decision Letter · Decision Letter 0]

28 May 2020

PONE-D-20-05121

Predisaster Predictors of Posttraumatic Stress Symptom Trajectories: An Analysis of Low-Income Women in the Aftermath of Hurricane Katrina

PLOS ONE

Dear Dr. Lowe,

Thank you for submitting your manuscript to PLOS ONE. After careful consideration, we feel that it has merit but does not fully meet PLOS ONE’s publication criteria as it currently stands. Therefore, we invite you to submit a revised version of the manuscript that addresses the points raised during the review process.

I have received three reviews and have read your interesting manuscript with pleasure. There is clearly a lack of true prospective studies in this field and your study, with very high response rates, will help to fill this gap. In brief, the reviewers recommended revisions and I agree with most of the comments. The clear comments of the reviewers are listed below and I therefore try not repeat them as much as possible in my comments that are listed below (L=line in manuscript).

Introduction

You assessed trajectories of PTSS but did not really define when collected data can be used to assess trajectories; which time intervals are acceptable to assess trajectories? Please explain why an 8-year time interval between T2 and T3 is acceptable. The time interval between T1 and T2 strongly differs from the time interval between T2 and T3. Please explain how we can interpret findings when the time intervals are so different.

L20

In the abstract you mentioned “*Participants (N = 885) were from the Resilience in Survivors of Katrina Study and were assessed once prior to the hurricane, and three times thereafter, at approximately 1, 4, and 12 years postdisaster*”.  However, your respondents were enrolled in a community college intervention at baseline. This should be mentioned more clearly in the abstract and in the first lines of the discussion section where you summarize main findings since this is a relevant but very specific study sample.

L39

In the introduction you wrote “*a large body of research has linked higher postdisaster perceived social support to lower postdisaster PTSS* (1,2)”. Both are indeed associated, but several longitudinal studies show that the associations are rather complex. As you know several studies fail to show that social support protects against the development of PTSD-symptomatology (cf. Nickerson et al., 2018; Yap & Devilly, 2004). As Kaniasty and Norris as well as others have shown both social causation as well as social selection may play role. Negative support or the lack of support seem more important the amount of support. I believe that you should mention both causation and selection more clearly in your introduction.

L47-L53

Readers may interpret this section as “*longitudinal studies assessing trajectories of PTSD are not capable to capture varied patterns*” while this is incorrect (cf. delayed onset PTSD, chronic PTSD). Please clarify what you want to say here a bit more in detail.

L71

In addition, following the meta-analyses of Galatzer et al, you concluded “*This suggests that there is selection bias in cohorts recruited in the aftermath of trauma, such that those suffering from mental health symptoms are likely to be overrepresented*”.  You may be right, but based on the arguments you provide the opposite can also be true, i.e. that in prospective studies (where the first pre-disaster survey is not trauma related) people with mental health problems participate less.

L153

Please explain briefly why you assesses the associations with and without adjustment for indicators of disaster exposure.

L156

Do you have any information about the proportion of people with the same characteristics as your study sample among the total groups affected adults? For which subgroup(s) are the results representative?

L157

Please clarify the intervention more in detail, and why you did not control for/take into account the possible effects of this intervention.

L215

According to your measures section you consider receiving benefits such as food stamp as a demographic characteristic. I believe it is much more appropriate to consider the use of for instance food stamps as an indicator for a lack of resources and/or vulnerability.

L265-281

I fully realize that there are no one-size-fits-all criteria for making decisions about which model should be chosen, but the results of your latent class analyses (Table 2) raise serious questions, given the provided info in the table and text in the results section, about if a 3-class solution is indeed the best solution. There is a serious drop in AIC, BIC and sample size adjusted BIC values after the 3-class solution, and an increase in entropy R. The 6-class solution does still improve the model. In addition, the tables does not provide information about the numbers (only the smallest class) in each class after each step. In other words, your table and statistics raise the question if a x-class solution (x >6) better represents the data than a 3-class solution. I therefore invite you to present all results of 7- to 10-classes latent class analyses (including the numbers of each class) to be better able to interpret your findings (and decisions).

L287 Table 3 and related text

You computed the means of all items of the IES-R and presented the mean scores in Table 3. Since many papers using the IES-R presented the total scores making comparison I ask you, to help readers who want to compare your finding with the findings of other studies, to present the total scores (and related sd’s). These crude means may further help the reader to understand why you label the trajectories as you did (why is a mean score of 2 high; high compared to what?). In addition, it would help the reader if you compute the prevalence of probable PTSD among the identified classes for the three waves using cut-off scores.

L 289 Figure 3 and related text

Please add 95% confidence intervals of the identified classes, since it may help the reader to understand why, although the difference between the crude means at T1 and T3 of class 3 and class 2 are almost similar, the slope of class 3 is not significant. Furthermore, the time between T2 and T3 was about 8 years. Given this very long period, I do not believe it is justified to connect the scores on the surveys with lines because it suggests something (scores in the years between) you in fact do not know.

L307 an L311 Table 4

Please add numbers of respondents of 3 classes in headings to help the reader (same for following tables). I would like to suggest to re-number “Table 4 continued” in Table 5 (and re-number other tables) because they are separate tables, and rename Table 4 in “*Descriptive Statistics for Participants with Most Likely Membership in Posttraumatic Stress Symptom Trajectory*”.

Table 4 and 5

I had some difficulties comparing the result of the univariate OR, model 1 and model 2 of the High-Decreasing vs. Moderate-Decreasing, High-Stable vs. Moderate-Decreasing, and High-Stable vs. High-Decreasing because I was forced to read the results of tables back and forth. I was wondering if you could re-arrange the results in the three tables in such a way that Table 5 shows the ORs of the univariate (model 1), multivariate (except disaster exposure, model 2) and full multivariate (model 3) analyses of High-Decreasing vs. Moderate-Decreasing comparison. In a similar way Table 6 showing the ORs of model 1, 2 and 3 with respect to High-Stable vs. Moderate-Decreasing etc.

L420

The time between the second and third survey was about 8 years, indicating that you do not have any data about PTSS of two third of the total study period. This is a very important limitation you do not pay any attention to. In addition, another important limitation is, given the total study period of 12 years, that you did not assess exposure to other potentially traumatic or stressful life events.

Discussion general

You did not really discuss the relevance of your findings for post-disaster mental health policies or interventions.

We look forward to receiving your revised manuscript.

Kind regards,

Peter G van der Velden, Ph.D.

Academic Editor

PLOS ONE

2. In the ethics statement in the Methods and online submission information, please clarify whether consent was written or verbal.  If verbal, please also specify: 1) whether the ethics committee approved the verbal consent procedure, 2) why written consent could not be obtained, and 3) how verbal consent was recorded. If the need for informed written consent was waived by the ethics committee, please include this information.

Reviewers' comments:

Reviewer's Responses to Questions

**Comments to the Author**

1. Is the manuscript technically sound, and do the data support the conclusions?

Reviewer #1: Yes

Reviewer #2: Yes

Reviewer #3: Yes

2. Has the statistical analysis been performed appropriately and rigorously? 

Reviewer #1: Yes

Reviewer #2: Yes

Reviewer #3: Yes

3. Have the authors made all data underlying the findings in their manuscript fully available?

Reviewer #1: Yes

Reviewer #2: Yes

Reviewer #3: Yes

4. Is the manuscript presented in an intelligible fashion and written in standard English?

Reviewer #1: Yes

Reviewer #2: Yes

Reviewer #3: Yes

5. Review Comments to the Author

Reviewer #1: This well-written manuscript has a number of noteworthy strengths, including its longitudinal examination that includes baseline assessment predisaster, and follow-up over 12 years, which would be a contribution to our understanding of postdisaster symptom trajectories. Although its focus on a primarily female, Black, low-income sample is not fully representative of community members who were exposed to Hurricane Katrina, this limitation was sufficiently addressed in the manuscript. I have identified areas in which additional details would be helpful to readers below.

Abstract

1. Page 2, line 21: Please specify the length of time prior to Hurricane Katrina that the baseline assessment was conducted.

2. Page 2, line 23: Please rephrase to clarify that each of the three factors were assessed predisaster. Also, how were physical conditions and problems distinguished?

3. Page 2, lines 26-32: If space permits, please present social support findings.

4. Page 2, line 36: If mental health problems were assessed predisaster, it may be more appropriate to replace 'prevent' with 'address' or other phrasing that more accurately places mental health problems in context postdisaster.

Background

5. Page 4, line 70: How was trauma severity defined, more specifically? Was this higher levels of exposure, for example?

6. Page 6, line 110: Please edit as 'postdisaster' (add 's').

7. Page 6, lines 120 & 122: Please be more specific regarding physical health here (e.g., physical symptoms, medical diagnoses, self-reported poor health?).

8. Page 6, line 128: Please consider mentioning protective factors here as well.

9. Page 6, line 129: Please provide more context regarding how disaster exposure is defined here. What would be the factors that would potentially result in increased exposure?

10. Please include a brief background regarding Hurricane Katrina, with the timeline presented.

Methods

11. Page 10, lines 198-200: Please provide a rationale for why health conditions and problems are being distinguished here. Is the distinction based on medical diagnosis?

12. Page 10, line 205: Please provide all 8 items included on the checklist that define disaster-related trauma exposure.

13. Page 11, line 227: Please consider adding 'year(s)' at each time point (i.e., "...at 1 year, T2 at 4 years, and T3 at 12 years...").

14. Page 12, lines 241-242: Please specify the resources and vulnerabilities included here.

Discussion

15. Page 27, line 413: Please specify: "...providing evidence that this type of exposure is only..." or something similar.

16. Please provide additional comment on the clinical implications over time, considering this rich dataset and the potential for additional postdisaster stressors to contribute to the trajectories over time.

Reviewer #2: From data collected over 12 years, the study identifies different trajectories of post-traumatic stress syndrome (PTSS) following Hurricane Katrina and the influence of predisaster social supports and mental and physical health within a sample of low-income women. While the sample is not representative of the broader population affected by Katrina, the study provides new insights about predisaster predictors of long-term PTSS outcomes which can guide targeted mental health service provision. The manuscript clearly lays out the rationale (with reference to current research evidence), methodological steps are appropriate and despite complex analysis, results are presented in a comprehensible way with meaningful discussion behind the identified PTSS typologies and study limitations. I have only a few comments for the authors’ consideration.

1. Participants were recruited for another study but subsequently dropped from that study – made me question the reason why and whether this reason has any consequence for outcomes in the current study?

2. From the manuscript, demographics were only collected at baseline – however changing characteristics such as less dependence on income support or change in relationship status may have also influenced PTSS trajectories. Can the authors comment on this and perhaps add to limitations section?

3. What was the rationale for choosing the particular health conditions? For e.g., why not Type 2 diabetes which is prevalent in low SES and Black populations?

4. Ethnicity was a significant factor in univariate and model 1 multivariable. What is the comparator group (i.e., what ethnicity made up the 13.8% non-Black participants)?

5. Was there any interaction analysis of co-morbidities, given frequent co-occurrence of mental and physical health issues? Similarly, with significant association of increasing age, could this be related also to having more children? (although perhaps this data is not available due to socio-demographics not collected at every timepoint?).

Reviewer #3: General

Very interesting study, well-conducted and empirically sound. It is highly significant that the authors had available pre-disaster data about their participants and that they had data of the mental health situation of 12 years after the disaster! Of course, there are some limitations to this study (e.g., the authors could have included other indicators (for example: gender) in their study).

Specifically

On page 3 the authors state that ‘the prevalence of postdisaster PTSD after disasters is estimated at 5-10% in the general population and 30-40% among direct victims’. This statement is puzzling. Most epidemiological studies on PTSD among disaster affected populations show a prevalence of 5 to 10 or 15 percent. On what studies is the rated of 30-40 percent based and what is the difference then with direct victims?

Interesting is the remark in the introduction on the noteworthy finding of Galatzer-Levy and colleagues’ review that the prevalence of participants in the resilience trajectory was significantly higher in prospective studies, relative to longitudinal studies, defined as studies with posttrauma data only. Unfortunately, the authors do not deal with that general finding any more in the discussion. Is this statement not relevant to their study?

Page 8. How characteristic were the RISK participants originally recruited as part of the Opening Doors Study for the New Orleans population? And why were they dropped from the larger study?

Page 8. Impressive response rate at Time 3!

Unlike most prior posttrauma trajectory studies, this study did not find a trajectory of consistently no/low PTSS (a resilient trajectory). This remains quite puzzling for me. Indeed, it could be a function of the sample characteristics, as the authors state. Yet, it is still very noteworthy, as most or nearly all trajectory studies have found that 65-80 percent of the people are in the resilience class. Please deal with this discrepancy a little bit more.

One could argue that the Moderate-Decreasing trajectory is just a variation of a resilient trajectory. Even if you were resilient directly after a disaster, your postdisaster responses (although not on a level of a mental disorder) slowly diminish in the course of time. For this argument one needs to know the mean score of the IES. For the moderate-decreasing trajectory this mean score was at the start of the study 1,22. That implies a sum score of nearly 27 on the IES-22. There are no specific cut-off score for this inventory, but generally scores higher than 24-33 are considered as of either concern or a sign of disorder. So, the term Moderate-Decreasing is justified, but nevertheless this is an issue for the discussion.

The authors should pay more attention in their discussion section to practical implications of their study. They only mention them in a couple of separate sentences (notably, the last sentence of the article).

In general, this is a fine study, but the discussion could use more academic reflection. What are theoretical implications of this study? Are there alternative interpretations and explanations? The discussion could go beyond a sober description of the results and implications for future research.

6. PLOS authors have the option to publish the peer review history of their article (what does this mean?). If published, this will include your full peer review and any attached files.

Reviewer #1: No

Reviewer #2: No

Reviewer #3: Yes: Rolf J. Kleber

---

## [Author Response · Author response to Decision Letter 0]

20 Aug 2020

(Please see attached document for formatting)

August 10, 2020

Peter G. van der Velden, Ph.D.

Academic Editor

PLOS ONE

Dear Dr. van der Velden:

My co-authors and I thank you for the opportunity to revise and resubmit our manuscript “Predisaster Predictors of Posttraumatic Stress Symptom Trajectories: An Analysis of Low-Income Women in the Aftermath of Hurricane Katrina” (Ms. No. PONE-D-20-05121) for publication consideration in PLOS ONE. We appreciate the Editor and Reviewers’ comments and have revised the manuscript accordingly. Key edits include: (1) inclusion of model with 7-10 latent classes; (2) enhanced efforts to situate the results of our study within prior research on posttrauma mental health trajectories; and (3) a more thorough discussion of the implications of our work for disaster preparedness and response. Below, we list each of the comments and how it was addressed in our revision.

Editor

1) Introduction: You assessed trajectories of PTSS but did not really define when collected data can be used to assess trajectories; which time intervals are acceptable to assess trajectories? Please explain why an 8-year time interval between T2 and T3 is acceptable. The time interval between T1 and T2 strongly differs from the time interval between T2 and T3. Please explain how we can interpret findings when the time intervals are so different.

The Editor raises an interesting point about what data are needed to assess developmental trajectories. It is our view that there are a variety of ways to use longitudinal data that are theoretically interesting and could have important clinical implications. For example, we are familiar with PTSS trajectories studies conducted using daily diary data in the first 5-6 months post-trauma (Hruska et al., 2016, Psychological Trauma) as well as over a 15-day period (Biggs et al., 2019, BMC Psychiatry). We believe that such data are useful in identifying survivors in need of services in the immediate aftermath of trauma and shedding light on factors contributing to daily fluctuations in PTSS, respectively, but potentially less so in understanding different pathways that might only emerge over a longer period of time. On the other hand, we believe that trajectories that cover a longer period of time would be useful in understanding the long-term impact of trauma and identifying factors shaping risk for chronically elevated symptoms or trajectories characterized by slow rates of recovery. More generally, trajectory studies that cover long periods of time shed light onto experiences and risk and resilience factors over the life course.

For the purposes of this revision, we delved into the trajectory literature, both with regard to posttrauma mental health symptoms and other indicators of functioning. It does seem like most posttrauma trajectory studies cover a shorter period of time than ours and, consequently, have narrower gaps between assessments. For example, in the Galatzer-Levy et al. (2018) review, the studies covering the longest periods of time both used data from the World Trade Center responder cohort, with data collected 3 years, 6 years, and 8 years after the September 11, 2001 terrorist attacks (Maslow et al., 2015 in Journal of Traumatic Stress; Pietrzak et al., 2014 in Psychological Medicine). In our own search, we identified a posttrauma trajectory study that covered a longer period of time (Holgersen et al., 2011, in Journal of Traumatic Stress) and that, interestingly, examined PTSS trajectories over both 6 time points in the first weeks posttrauma, as well as general mental health trajectories 4 time points at baseline, 1 year, 5 years, and 27 years posttrauma – thus including an even wider gap between the final two assessments than in our study. In this manuscript, the authors emphasized a key strength of their study – its long-term follow-up – and they argued for additional studies that follow disaster survivors for decades postdisaster. As such, although the Reviewer is correct that the wide gap between our assessments is a limitation – and ideally, we would have liked to re-interview our participants more often over the study period! – this should be considered alongside the strengths of long-term postdisaster data, high retention rates (as noted by Reviewer 3), a fairly large sample size, and inclusion of at-risk adults.

We also note here that we identified trajectory studies of other health conditions that spanned long periods of time and included wide gaps in assessment, including depression (Cronkite et al., 2013 in Journal of Affective Disorders; time points: baseline, 1 year, 4 years, 10 years, and 23 years), substance use (Chen & Jacobson, 2012 in Journal of Adolescent Health; time points: baseline, 1 year, 7 years, 13 years), and lung function (Bui et al., 2018 in Lancet Respiratory Medicine; time points: age 7, age 13, age 18, age 45, age 50, and age 53). Thus, while most trajectory studies seem to cover a shorter period of time and include narrower gaps between assessment periods, our study is consistent with other work on PTSS and other health conditions.

Nonetheless, while we contend that our data are suitable to conduct a trajectory analysis, we agree that the wide gaps between time points is a key limitation of our study that we overlooked in our initial draft. We have thus added the following to our limitations section: “Fourth, the time lag between the different study waves varied considerably, and eight years had passed between the second and third postdisaster assessments. The gaps between assessments mean that we do not have a complete picture of PTSS trajectories in the sample over the study period. It is possible that we would have identified a different set of PTSS trajectories, with distinctive patterns and predictors, if we had access to additional waves of data within the study timeframe” (pp. 36-37).

2) L20: In the abstract you mentioned “Participants (N = 885) were from the Resilience in Survivors of Katrina Study and were assessed once prior to the hurricane, and three times thereafter, at approximately 1, 4, and 12 years postdisaster”. However, your respondents were enrolled in a community college intervention at baseline. This should be mentioned more clearly in the abstract and in the first lines of the discussion section where you summarize main findings since this is a relevant but very specific study sample.

As requested, the revised manuscript mentions that participants were recruited as part of a community college intervention study both in the Abstract (p. 2) and the first paragraph of the Discussion (pp. 31-32).

3) L39: In the introduction you wrote “a large body of research has linked higher postdisaster perceived social support to lower postdisaster PTSS (1,2)”. Both are indeed associated, but several longitudinal studies show that the associations are rather complex. As you know several studies fail to show that social support protects against the development of PTSD-symptomatology (cf. Nickerson et al., 2018; Yap & Devilly, 2004). As Kaniasty and Norris as well as others have shown both social causation as well as social selection may play role. Negative support or the lack of support seem more important the amount of support. I believe that you should mention both causation and selection more clearly in your introduction.

We agree with the Editor that relationships between postdisaster perceived social support and PTSS are more complex than conveyed in our original submission. As suggested, we have therefore mentioned social causation and social selection in our revision, and have cited studies demonstrating the complexity of these processes in the aftermath of disasters:

“Pathways from perceived social support and mental health symptoms, including PTSS, are thought to be bidirectional. On the one hand, consistent with social causation, low social support could increase vulnerability to symptoms; on the other, consistent with social selection, symptoms could impede the development and maintenance of supportive social relationships or lead one to perceive their relationships as less supportive (Dohrenwend, 2000). Both mechanisms have received empirical support in the aftermath of disasters, with patterns of results varying depending on the nature of support provided and timing of assessment (Kaniasty & Norris, 2008; Platt et al., 2016).” (p. 6; cited references specified here for clarity).

In response to this comment, as well as Reviewer 3’s Comment #9 requesting more academic reflection on the results, we also returned to the issue of social causation and social selection in the Discussion: 

“…it is possible that the role of predisaster perceived social support in shaping risk for short-term elevations in PTSS is via other predisaster factors, particularly predisaster mental health problems – since predisaster probable mental illness remained significantly associated with the High-Decreasing trajectory, versus the Moderate-Decreasing trajectory, in the model. A pathway from predisaster perceived social support to postdisaster PTSS via predisaster mental illness would be consistent with social causation. On the other hand, it is also possible that our pattern of results is indicative of social selection – if, for example, predisaster perceived social support and PTSS trajectories were only related in univariate models because they shared predisaster mental illness as a common predictor. Future prospective studies with multiple waves of predisaster data could shed additional light on this issue” (p. 32).

4) L47-L53: Readers may interpret this section as “longitudinal studies assessing trajectories of PTSD are not capable to capture varied patterns” while this is incorrect (cf. delayed onset PTSD, chronic PTSD). Please clarify what you want to say here a bit more in detail.

We are unsure as to whether we agree with this assessment, as the text from our original manuscript read, “cross-sectional research fails to capture varied patterns, or trajectories, of PTSS” and thus does not refer to longitudinal studies. Nonetheless, we have added the following text to the Introduction to clarify the advantage of longitudinal data in estimating PTSS trajectories:

“Although information on earlier levels of PTSS could be collected at a single time point, this would be inherently subject to retrospective bias. Longitudinal data, with multiple assessments of PTSS over time, overcomes this limitation and thus allows for more precise information on the prevalence and predictors of PTSS trajectories” (p. 4).

5) L71: In addition, following the meta-analyses of Galatzer et al, you concluded “This suggests that there is selection bias in cohorts recruited in the aftermath of trauma, such that those suffering from mental health symptoms are likely to be overrepresented”. You may be right, but based on the arguments you provide the opposite can also be true, i.e. that in prospective studies (where the first pre-disaster survey is not trauma related) people with mental health problems participate less.

We agree with this assessment and have revised this statement so that it is clear that it was the authors of the meta-analysis who concluded that there was likely selection bias in cohorts recruited in the aftermath of disasters (p. 5). We have also added the opposite interpretation, as recommended: “An alternative explanation would be the opposite, i.e., that those suffering from mental health symptoms are underrepresented in prospective studies” (p. 5).

6) L153: Please explain briefly why you assesses the associations with and without adjustment for indicators of disaster exposure.

We have clarified this point in the Current Study paragraph. It now reads as follows: 

“We ran predictive models first without and then with adjustment for indicators of disaster exposure to assess whether relationships between predisaster factors and the odds of PTSS trajectories marked by symptoms elevations were attenuated in the latter as compared to the former. Such findings would suggest that the influence of predisaster factors on PTSS trajectory membership might be indirect via disaster exposure” (pp. 9-10).

7) L156: Do you have any information about the proportion of people with the same characteristics as your study sample among the total groups affected adults? For which subgroup(s) are the results representative?

This is a very good question, and we unfortunately do not believe we can come up with a good estimate of the proportion of New Orleans residents at the time of Hurricane Katrina who would fit the original study’s inclusion criteria – i.e., a student at one of two participating community colleges, a parent, between the ages of 18 and 34 years, and earning less than 200% of the poverty line. We considered using Census data to estimate the number and percentage of low-income parents in the New Orleans at the time of the hurricane. However, data are not available at this level of detail from prior to Katrina on the current U.S. Census Bureau data website (data.census.gov). Even if we were able to ascertain these values, however, they would not be reflective of our study sample, given that the participants were community college students at baseline.

We believe that we have addressed this limit to external validity sufficient in our Limitations section, in the following text:

“Fifth, although the low-income, predominantly Black, mothers in our sample were of interest given their risk for postdisaster adversity (Goldman & Galea, 2014), they are not representative of Katrina survivors from New Orleans or disaster survivors more generally. They also are not representative of all low-income mothers living in New Orleans at the time of Katrina, given that they were enrolled in a community college intervention at baseline” (p. 37; reference included here for clarity).

We have also updated the Method section to specify the inclusion criteria for the Opening Doors study, and that the majority of the New Orleans participants were low-income, non-Hispanic Black and unmarried mothers (p. 10).

8) L157: Please clarify the intervention more in detail, and why you did not control for/take into account the possible effects of this intervention.

We have provided additional information about the intervention: “Opening Doors participants were randomly assigned to either an intervention condition, in which they received extra advising and a $1,000 stipend for each subsequent semester, or a control condition, in which they received neither of these benefits” (p. 10).

In addition, we note that no significant differences in postdisaster mental health were observed between participants in the intervention and control conditions and, as such, intervention status was not included in the primary analyses of the RISK data nor in the current study (p. 10).

9) L215: According to your measures section you consider receiving benefits such as food stamp as a demographic characteristic. I believe it is much more appropriate to consider the use of for instance food stamps as an indicator for a lack of resources and/or vulnerability.

We have considered this suggestion and have opted to retain the number of benefits as a demographic covariate. This covariate was included to be an indicator of socioeconomic status, which is typically included as a demographic characteristic in disaster research (c.f., Galea et al., 2007; Tracy et al., 2011). Additionally, we were unsure as to whether the receipt of benefits would be best conceptualized as a predisaster vulnerability or resource. Certainly, low socioeconomic status, of which we consider receipt of benefits an indicator, would be conceptualized as a vulnerability factor. On the other hand, not all people with low income qualify for or access benefits, and the receipt of benefits can certainly be a resource for those who do.

10) L265-281: I fully realize that there are no one-size-fits-all criteria for making decisions about which model should be chosen, but the results of your latent class analyses (Table 2) raise serious questions, given the provided info in the table and text in the results section, about if a 3-class solution is indeed the best solution. There is a serious drop in AIC, BIC and sample size adjusted BIC values after the 3-class solution, and an increase in entropy R. The 6-class solution does still improve the model. In addition, the tables does not provide information about the numbers (only the smallest class) in each class after each step. In other words, your table and statistics raise the question if a x-class solution (x >6) better represents the data than a 3-class solution. I therefore invite you to present all results of 7- to 10-classes latent class analyses (including the numbers of each class) to be better able to interpret your findings (and decisions).

The Editor is correct that the model selection for the current study was not straightforward and that we had to simultaneously take various statistical and theoretical criteria into account when making our decision. We made efforts to be explicit in the myriad factors informing our decision in the text, for example noting our observations regarding the small number of participants in the smallest class for solutions with five or more classes; the observed “leveling-off” of the AIC, BIC, and adjusted BIC values; and inspection of plots of estimated trajectories. Upon reviewing the text, we believe that we gave sufficient detail regarding our decision, while inviting interested readers to request additional information (p. 17).

We had uncertainty regarding the Editor’s suggestion to run models with 7-10 classes. We had initially limited ourselves to six classes, as we are aware of only a few PTSS LCGA studies that have this many classes, with most studies reporting an optimal solution with either three or four classes. Additionally, we had already noted that the 5- and 6-class solutions from our data had a smallest class with only 1.0% of the sample (n = 9) and were doubtful that solutions with more classes would resolve this issue. Nonetheless, we have opted to run the additional models for our revision and have included the results (pp. 14, 17, 19-21). We note here that those solutions also had smallest classes with very few participants, ranging from 9 to 11 (1.0-1.2% of the sample) and were excluded from consideration for this reason (p. 17).

In our revision, we have also included the number and percentage of participants with most likely membership in each class for each solution in our table (p. 20-21). 

11) L287 Table 3 and related text: You computed the means of all items of the IES-R and presented the mean scores in Table 3. Since many papers using the IES-R presented the total scores making comparison I ask you, to help readers who want to compare your finding with the findings of other studies, to present the total scores (and related sd’s). These crude means may further help the reader to understand why you label the trajectories as you did (why is a mean score of 2 high; high compared to what?). In addition, it would help the reader if you compute the prevalence of probable PTSD among the identified classes for the three waves using cut-off scores.

We appreciate this suggestion, as we are aware that many studies use the summed item scores and probable PTSD cut-offs for the IES-R. Our team had actually done this ourselves in prior work, but were then informed by one of the original scale authors (Daniel S. Weiss, Ph.D.) that both were inappropriate uses of the scale. We have appended to this letter text from a document that Dr. Weiss sent us that speaks to both issues, with relevant text highlighted. Because we have received this feedback from the scale author, it is our preference to solely report mean item scores. To address this point, we have revised our description of the IES-R to include that scale scores were computed as the mean of all items “in accordance with the scale instructions” (p. 11). However, if the Editor and/or Reviewers feel strongly on this matter, we are willing to provide either/both the summed scores and percentages exceeding the 1.5 cutoff (equivalent to a summed score of 33) for probable PTSD.

Our naming of trajectories was also informed by Dr. Weiss’s document. For example, he wrote “… if an individual’s score or a group’s mean on the Intrusion subscale was 1.89, that would indicate that for intrusion, for this person (group), in the last week their distress from intrusive symptoms was close to, but not quite moderate.” We made two changes to our manuscript to make this more explicit. First, we listed all response options to the IES-R to our description of the measure on page 11. Second, the sentence describing how trajectories were named now reads: “Trajectories were named based on descriptive results, specifically the correspondence between mean PTSS scores for participants with most likely membership in the given trajectory at each time point and IES-R item ratings, and the statistical significance of growth terms” (p. 23). We note here that we considered scores between 2 and 3 “High” because they would indicate that the participant was bothered by PTS symptoms, on average, more often than “moderately” – that is, comparisons were made to other possible scores on the scale. We considered the term “High-Moderate” to reflect that these scores were not at the ceiling of the scale’s range, but felt that that would be cumbersome to readers. However, we are willing to make this change, or consider alternative labels if the Editor and/or Reviewers have specific suggestions.

12) L 289 Figure 3 and related text: Please add 95% confidence intervals of the identified classes, since it may help the reader to understand why, although the difference between the crude means at T1 and T3 of class 3 and class 2 are almost similar, the slope of class 3 is not significant. Furthermore, the time between T2 and T3 was about 8 years. Given this very long period, I do not believe it is justified to connect the scores on the surveys with lines because it suggests something (scores in the years between) you in fact do not know.

We appreciate the Editor’s thoughtful consideration about the results of our study. We have edited both Table 3 and the text to add 95% confidence intervals (CIs) for the growth terms and estimated means of PTSS, which we agree will help readers recognize why the slope of the High-Decreasing trajectory reached statistical significance, whereas that of the High-Stable trajectory did not (pp. 22-23). The PTSS values reported in Table 3 are notably slightly different from those reported in our original submission, which were based in the observed sample data and thus included only the subsamples of cases who participated at any given wave. Although the means that we now report in Time 3 are thus duplicative of those plotted in Figure 1, we believe that they will be more useful to readers as they reflect estimated values for the full analytic sample and that the standard errors and 95% CIs will help readers interpret the results.

We also created a version of the Figure with 95% CIs around estimated means, which we have pasted below. However, we have opted not to include this version since we believe that the overlap between the 95% CIs for the High-Stable and High-Decreasing trajectories at Times 1 and 2 make it somewhat difficult to interpret and we are providing the 95% CIs for readers who are interested in seeing them in Table 3.

Regarding the connecting lines in the Figure, we have opted to retain them for two reasons. First, we are unaware of any trajectory studies that present plots without connecting lines, and we would prefer to be consistent with prior work. Second, although we considered omitting lines and instead using different symbols to identify the three trajectories, we believe that the classes are best distinguished visually using connecting lines. Nonetheless, we concur that it is worth emphasizing the gaps in assessment over the study period and hence our limited ability to capture variability in PTSS over time among the study participants. We have therefore opted to retain connecting lines on the figure, but have edited them so that they are all dotted and dashed, rather than solid lines. Visually speaking, we believe that this change better reflects uncertainty in participants’ PTSS between study periods. Additionally, we have added the following text to our description of the Figure: “We note here that we have included dashed or dotted lines to connect data points over time for each trajectory to illustrate changes in PTSS across the three postdisaster waves. It is worth emphasizing, however, that this depiction does not capture the likely non-linear changes in PTSS between waves, which would require additional data points over the study period” (p. 18). As mentioned previously, we have also added the gaps between assessments, particularly between the second and third postdisaster waves, to the Limitation section (pp. 36-37). 

We also considered omitting the figure altogether in response to the Editor’s concern and also because the estimated means are now listed in Table 3. However, we believe that a visual depiction of the findings will help readers interpret our findings and is consistent with how trajectories are presented in most other work. That said, if the Editor would like us to omit the figure to concerns about it being misleading (or replace the current figure with the one that includes 95% confidence intervals) due, we are willing to do so.

13) L307 an L311 Table 4: Please add numbers of respondents of 3 classes in headings to help the reader (same for following tables). I would like to suggest to re-number “Table 4 continued” in Table 5 (and re-number other tables) because they are separate tables, and rename Table 4 in “Descriptive Statistics for Participants with Most Likely Membership in Posttraumatic Stress Symptom Trajectory”.

As suggested, we have added the number of respondents in the three classes to Tables 4 and 5 (pp. 24, 27-29). We agree that this information would be beneficial to readers in their interpretation of the results.

We have also removed the univariate analysis results from Table 4, as described in more detail in our response to the following comment.

14) Table 4 and 5: I had some difficulties comparing the result of the univariate OR, model 1 and model 2 of the High-Decreasing vs. Moderate-Decreasing, High-Stable vs. Moderate-Decreasing, and High-Stable vs. High-Decreasing because I was forced to read the results of tables back and forth. I was wondering if you could re-arrange the results in the three tables in such a way that Table 5 shows the ORs of the univariate (model 1), multivariate (except disaster exposure, model 2) and full multivariate (model 3) analyses of High-Decreasing vs. Moderate-Decreasing comparison. In a similar way Table 6 showing the ORs of model 1, 2 and 3 with respect to High-Stable vs. Moderate-Decreasing etc.

We thank the Editor for this suggestion and agree that presenting the results of the univariate analyses alongside the multivariable models makes good sense. We have therefore edited Table 5 to include the results of the univariate analyses (pp. 27-29). It is our preference to maintain Table 5 as a single table, to reflect that all between-class trajectories were conducted within single analyses. However, we are willing to break this into three tables, as the Editor suggests, if need be. 

We note here that, in reviewing the univariate results of our original submission, we observed that one of the estimates (for predisaster physical health conditions or problems as predictive of High-Decreasing versus Moderate-Decreasing) was non-significant, yet its confidence interval did not include 1.00. Based on information on the Mplus website regarding such inconsistent results, we re-ran the LCGA and predictive analyses using 1,000 bootstrapped samples (p. 15). The only change in the results with this alteration was for the High-Decreasing versus Moderate-Decreasing comparison for predisaster physical health conditions or problems in the univariate analysis.

15) L420: The time between the second and third survey was about 8 years, indicating that you do not have any data about PTSS of two third of the total study period. This is a very important limitation you do not pay any attention to. In addition, another important limitation is, given the total study period of 12 years, that you did not assess exposure to other potentially traumatic or stressful life events.

We thank this Reviewer for noting the gaps between study waves, and in particular the eight-year gap between the second and third post-disaster waves. We agree that gaps in assessment preclude us from presenting a complete picture of participants’ PTSS trajectories over the study period. We have therefore added the following to the Limitations section:

“Fourth, the time lag between the different study waves varied considerably, and eight years had passed between the second and third postdisaster assessments. The gaps between assessments mean that we do not have a complete picture of PTSS trajectories in the sample over the study period. It is possible that we would have identified a different set of PTSS trajectories, with distinctive patterns and predictors, if we had access to additional waves of data within the study timeframe” (pp. 36-37). 

Despite this key limitation, we believe that our study has value given its other strengths, including access to predisaster data, focus on a sample of at-risk adults, and relatively large sample size.

We also concur that another important limitation to the analysis is that we did not include exposure to other potentially traumatic or stressful life event in our predictive models. However, we believe that we had adequately covered this point in our original submission. First, in the Discussion section, we state the possibility the intervening trauma and stressor exposure distinguish between survivors who do and do not recover (p. 35). We have updated this statement to point out that we did not include subsequent exposures in the analysis (p. 35). Second, in the Limitations paragraph, we list prior exposure to traumatic and stressful life events as important predisaster factor to include in future studies, and cite prior research showing associations between predisaster exposures and postdisaster mental health (p. 36).

16) Discussion general: You did not really discuss the relevance of your findings for post-disaster mental health policies or interventions.

We agree with this assessment and have added two paragraphs on the implications of our findings for disaster preparedness and response (pp. 35-36).

Reviewer 1

1) This well-written manuscript has a number of noteworthy strengths, including its longitudinal examination that includes baseline assessment predisaster, and follow-up over 12 years, which would be a contribution to our understanding of postdisaster symptom trajectories. Although its focus on a primarily female, Black, low-income sample is not fully representative of community members who were exposed to Hurricane Katrina, this limitation was sufficiently addressed in the manuscript. I have identified areas in which additional details would be helpful to readers below.

We thank this Reviewer for the thoughtful comments and attention to both the strengths and limitations of our work.

2) Abstract: Page 2, line 21: Please specify the length of time prior to Hurricane Katrina that the baseline assessment was conducted.

The Abstract has been updated to specify that baseline assessments were conducted 6-21 months predisaster (p. 2).

3) Abstract: Page 2, line 23: Please rephrase to clarify that each of the three factors were assessed predisaster. Also, how were physical conditions and problems distinguished?

As suggested, the abstract specifies that each of the three factors were assessed predisaster (p. 2).

In addition, we provided more detail regarding the assessment of physical health conditions and problems, which we hope will shed light on how they were distinguished. Specifically, physical health conditions reflected lifetime medical diagnoses, whereas physical health problems reflected symptoms that participants were experiencing at the time of the hurricane (p. 12). We note here that, upon review of the text, we noticed that we had inadvertently omitted two of the physical health conditions in our original draft. Specifically, participants were asked whether they had ever received a diagnosis of diabetes, and whether they had received a diagnosis of any other medical condition. These are included in the revised manuscript (p. 12).

4) Abstract: Page 2, lines 26-32: If space permits, please present social support findings.

We have added the perceived social support findings to the Abstract, while staying with in the 300-word limit (p. 2).

5) Abstract: Page 2, line 36: If mental health problems were assessed predisaster, it may be more appropriate to replace 'prevent' with 'address' or other phrasing that more accurately places mental health problems in context postdisaster.

We have removed the statement in question from the Abstract.

6) Background: Page 4, line 70: How was trauma severity defined, more specifically? Was this higher levels of exposure, for example?

We thank this Reviewer for drawing our attention to the statement regarding trauma severity in the Galatzer-Levy et al. (2018) meta-analysis. Upon re-review of this study, we realized that it did not systematically assess the severity of trauma exposure as a predictor of trajectory membership. We have therefore opted to omit this sentence entirely from our revised version.

7) Background: Page 6, line 110: Please edit as 'postdisaster' (add 's').

We have made this correction and thank this Reviewer for the attention to detail (p. 7).

8) Background: Page 6, lines 120 & 122: Please be more specific regarding physical health here (e.g., physical symptoms, medical diagnoses, self-reported poor health?).

As requested, we have included the specific physical health indices included in prior research (p. 8).

9) Background: Page 6, line 128: Please consider mentioning protective factors here as well.

As suggested, we have included protective factors in this statement (p. 8).

10) Background: Page 6, line 129: Please provide more context regarding how disaster exposure is defined here. What would be the factors that would potentially result in increased exposure?

We have provided the following background information regarding disaster exposure: “The ways in which disaster exposure is measured varies considerably across studies, but various indicators, including counts of disaster-related trauma (e.g., limited access to life-sustaining resources, perceived life threat) and specific experiences (e.g., bereavement, property damage), have been linked to adverse postdisaster mental health outcomes (Goldmann & Galea, 2014; Lowe, Bonumwezi, Valdespino-Hayden, & Galea, 2018)” (p. 8, references specified here for clarity). In addition, we have specified that the prior analyses of RISK data cited used a count of disaster-related trauma as the indicator of exposure (p. 8).

11) Background: Please include a brief background regarding Hurricane Katrina, with the timeline presented.

As requested, we have provided some basic information regarding Hurricane Katrina, including the date it made landfall in the Gulf Coast, and the number of deaths and estimated costs of damages attributed to the disaster (p. 9). We have also included the timeline of the study in the Current Study section (p. 9).

12) Methods: Page 10, lines 198-200: Please provide a rationale for why health conditions and problems are being distinguished here. Is the distinction based on medical diagnosis?

We thank this Reviewer for the clarification request. As specified in our response to this Reviewer’s Comment #3, health conditions were medical diagnoses that the participant had received in their lifetime, and problems were symptoms the participant was currently experiencing. We have revised the text to clarify this (p. 12).

13) Methods: Page 10, line 205: Please provide all 8 items included on the checklist that define disaster-related trauma exposure.

As requested, we have included all 8 items from the checklist of disaster-related trauma (p. 13).

14) Methods: Page 11, line 227: Please consider adding 'year(s)' at each time point (i.e., "...at 1 year, T2 at 4 years, and T3 at 12 years...").

We considered this suggestion, but opted to omit “years” as the goal of this statement was to highlight the numeric values used in the trajectory analysis. However, we added “years” when we mentioned the timing of assessments at various other points in the manuscript based on this comment (e.g., pp. 2, 9, 30).

15) Methods: Page 12, lines 241-242: Please specify the resources and vulnerabilities included here.

As requested, we have listed the predisaster resources and vulnerabilities included in the model (p. 14). For consistency, we have also specified which demographic characteristics and indicators of disaster exposure were included in predictive models in the Data Analysis section as well (pp. 14-15).

16) Discussion: Page 27, line 413: Please specify: "...providing evidence that this type of exposure is only..." or something similar.

We have edited this sentence per this Reviewer’s suggestion (p. 35).

17) Discussion: Please provide additional comment on the clinical implications over time, considering this rich dataset and the potential for additional postdisaster stressors to contribute to the trajectories over time.

In the revised manuscript, we have included two paragraphs on the implications of the study for disaster preparedness and response (pp. 35-36). In doing so, we suggested that clinicians should be attuned to participants’ exposure to other trauma and stressors over the life-course, consistent with the recommendations for trauma informed care (p. 36). We also mention the potential for other postdisaster trauma and stressors to contribute to trajectories in our interpretation of results (p. 35).

Reviewer 2

1) From data collected over 12 years, the study identifies different trajectories of post-traumatic stress syndrome (PTSS) following Hurricane Katrina and the influence of predisaster social supports and mental and physical health within a sample of low-income women. While the sample is not representative of the broader population affected by Katrina, the study provides new insights about predisaster predictors of long-term PTSS outcomes which can guide targeted mental health service provision. The manuscript clearly lays out the rationale (with reference to current research evidence), methodological steps are appropriate and despite complex analysis, results are presented in a comprehensible way with meaningful discussion behind the identified PTSS typologies and study limitations. I have only a few comments for the authors’ consideration.

We thank this Reviewer for the positive comments about our work, and useful suggestions to improve our manuscript.

2) Participants were recruited for another study but subsequently dropped from that study – made me question the reason why and whether this reason has any consequence for outcomes in the current study?

We have added to the manuscript that, because both community colleges were closed for the Fall 2005 semester due to Hurricane Katrina, the New Orleans participants were dropped from the larger Opening Doors study (p. 10). We note that no differences in postdisaster mental health outcomes were observed between participants in the intervention and control conditions and, following the primary analyses of the RISK data, we therefore did not include intervention status as a covariate in the current study (p. 10).

The Reviewer raises an interesting question as to whether the closing of the community colleges had impacts on the outcomes in the current study. Certainly, school closures would disrupt the participants’ academic careers and potentially their social networks as well, but perhaps not above and beyond other hurricane-related impacts in the New Orleans area. Regardless, we did not quantitatively assess participants’ perceived impact of school closures and thus could not include this in the analysis. 

3) From the manuscript, demographics were only collected at baseline – however changing characteristics such as less dependence on income support or change in relationship status may have also influenced PTSS trajectories. Can the authors comment on this and perhaps add to limitations section?

We agree that changes in demographic characteristics, as well as other time-varying risk and protective factors, could influence trajectory membership. However, the focus of this analysis was on the role of predisaster factors, rather than changes over time, in predicting trajectories. Nonetheless, especially since we agree with this Reviewer’s assessment, we have added the following statement to the Discussion: “Future studies could also include changes in general psychological distress, physical health and perceived social support, as well as in demographic characteristics, including participants’ access to social benefits, marital and cohabitation status, and number of children, which could also potentially shape the odds of recovery from initially elevated postdisaster symptoms” (p. 35).

4) What was the rationale for choosing the particular health conditions? For e.g., why not Type 2 diabetes which is prevalent in low SES and Black populations?

As noted in our response to Reviewer 1’s Comment #3, when working on our revision, we noticed that we had failed to list two of the items on our list of physical health conditions – diabetes, and any other health condition. These are now included among the list of conditions on page 12. We note here that we included all physical health conditions and problems that were assessed in the original Opening Doors study. As we were not involved in that initial study, we are unsure why the particular conditions and symptoms were selected for inclusion, and could not find any information from the Opening Doors study website. If this Reviewer feels strongly about this issue, we are willing investigate further. 

5) Ethnicity was a significant factor in univariate and model 1 multivariable. What is the comparator group (i.e., what ethnicity made up the 13.8% non-Black participants)?

We have provided more detailed information on non-Black participants on page 13: “…of the 855 participants who reported on race/ethnicity at baseline, 85 (9.9%) identified as non-Hispanic white, 20 (2.3%) as Hispanic, and 14 (1.6%) as other race/ethnicity. Because of these small subsample sizes, especially of the latter two groups, these categories were combined into the larger “Other” classification.” 

6) Was there any interaction analysis of co-morbidities, given frequent co-occurrence of mental and physical health issues? Similarly, with significant association of increasing age, could this be related also to having more children? (although perhaps this data is not available due to socio-demographics not collected at every timepoint?).

This Reviewer brings up some interesting suggestions for future research. However, we believe that these are both beyond the scope of the current study. First, we agree that it would be interesting to assess whether different predisaster comorbidities are predictive of PTSS trajectories. However, the purpose of this study was to assess the main effects of the three predisaster factors of interest. Our aims were based on the findings of prior research, and no studies to our knowledge have examined interactions of predisaster mental and physical health comorbidities as predictive of postdisaster mental health outcomes. We have nonetheless added that future longitudinal studies could examine interactions between various predisaster resources and vulnerabilities as predictors of PTSS trajectory membership (p. 33). 

Second, we agree that it would be interesting to examine whether changes in demographic characteristics influenced the odds of different trajectories, and we look forward to doing this in future research. However, given that the focus of this paper was on predisaster resources and vulnerabilities, we opted not to pursue this line of research in the current paper. We have nonetheless added changing demographics as another potential factor that could distinguish between the High-Stable and High-Decreasing trajectories (p. 35). 

Reviewer 3

1) Very interesting study, well-conducted and empirically sound. It is highly significant that the authors had available pre-disaster data about their participants and that they had data of the mental health situation of 12 years after the disaster! Of course, there are some limitations to this study (e.g., the authors could have included other indicators (for example: gender) in their study).

We thank this Reviewer for both the positive appraisal of our work and useful suggestions for revision.

2) On page 3 the authors state that ‘the prevalence of postdisaster PTSD after disasters is estimated at 5-10% in the general population and 30-40% among direct victims’. This statement is puzzling. Most epidemiological studies on PTSD among disaster affected populations show a prevalence of 5 to 10 or 15 percent. On what studies is the rated of 30-40 percent based and what is the difference then with direct victims?

We thank this Reviewer for pointing out this issue. We went back to the source of the statement, which was a review of the disaster mental health literature, which did not clarify the difference between the general population affected and direct victims. We have therefore opted to edit this statement so that it only refers to the general population (p. 4). 

3) Interesting is the remark in the introduction on the noteworthy finding of Galatzer-Levy and colleagues’ review that the prevalence of participants in the resilience trajectory was significantly higher in prospective studies, relative to longitudinal studies, defined as studies with posttrauma data only. Unfortunately, the authors do not deal with that general finding any more in the discussion. Is this statement not relevant to their study?

We thank this Reviewer for drawing our attention to this statement and how it might relate to our findings. In our initial draft, we had included this statement simply to illustrate the value of prospective studies such as ours. Based on the Editor and this Reviewer’s feedback, however, in our revision, we more carefully considered the relevance of this finding to our results. As detailed in our response to this Reviewer’s Comment #6, we believe that our finding could indicate that the opposite of Galatzer-Levey and colleagues’ interpretation could be true – i.e., that longitudinal studies (defined as posttrauma-only studies) could be overestimating the prevalence of resilience, that those dealing with severe posttrauma stressors might be less likely than their counterparts to enroll in new research studies, or that the lower proportions of resilient participants they observed in prospective studies could be an artifact of how symptoms were assessed – i.e., not in reference to a focal trauma. We have made a statement to this end on page 34. Given the alternative explanation for our finding (i.e., that the lack of a resilient trajectory was due to the sample characteristics), we were careful to state that additional research, particularly prospective studies that concurrently examine pre to posttrauma trajectories of symptoms not tied to a focal trauma (e.g., depression, general psychological distress) and posttrauma-only PTSS trajectories are needed to shed light on this issue (p. 34).

4) Page 8. How characteristic were the RISK participants originally recruited as part of the Opening Doors Study for the New Orleans population? And why were they dropped from the larger study?

We have updated the manuscript to provide inclusion criteria for the Opening Doors study, namely that students at participating community colleges had to be a parent, between the ages of 18 and 34 years old, and earning less than 200% of the federal poverty line (p. 10). As stated in the Discussion (p. 37), we do not consider the sample representative of Katrina survivors from New Orleans. Additionally, given that they were all community college students at baseline, we do not consider them representative of all low-income mothers living in New Orleans at the time of Katrina (p. 37). As noted in our response to the Editor’s Comment #7, we considered using Census data to estimate the number and percentage of low-income mothers in the New Orleans at the time of the hurricane. However, data are not available at this level of detail from prior to Katrina on the current U.S. Census Bureau data website (data.census.gov), and, even if we were to ascertain these values, they would not be reflective of our study sample, given that the participants were community college students at baseline.

The revised manuscript now specifies that participants were dropped from the Opening Doors study given that the two New Orleans community colleges were closed for the Fall 2005 semester (p. 10).

5) Page 8. Impressive response rate at Time 3!

We thank you for your attention to this strength of our study.

6) Unlike most prior posttrauma trajectory studies, this study did not find a trajectory of consistently no/low PTSS (a resilient trajectory). This remains quite puzzling for me. Indeed, it could be a function of the sample characteristics, as the authors state. Yet, it is still very noteworthy, as most or nearly all trajectory studies have found that 65-80 percent of the people are in the resilience class. Please deal with this discrepancy a little bit more.

We thank this Reviewer for the thoughtful comment, which made us more carefully consider how our study fits within the existing body of literature. As we noted in the Introduction, the only trauma-related trajectory studies with prospective data have identified trajectories of symptoms that are not tied to a given traumatic event. We know of no existing studies that have looked at PTSS trajectories in a sample similar to that of the RISK study – i.e., that looked at PTSS tied to an event that occurred over the course of the study. While we continue to believe that the lack of a resilient trajectory is likely due in large part to the sample characteristics, it is also possible that the pattern of results was due to the unique nature of our study – i.e., that we assessed predisaster predictors of PTSS using prospective data. In this sense, our findings, like the Galatzer-Levy et al. (2008) review suggest that posttrauma studies might be subject to selection bias, but in the opposite direction. 

We have addressed this comment in two ways. First, we state in the Introduction section that Galatzer-Levy and colleagues’ finding that rates of resilience were higher in what they define as prospective studies might have been due to how symptoms were assessed – namely, that they, logically, were not tied to a specific traumatic event (p. 5). We then specify that an alternative means of incorporating data would be to examine predisaster factors as predictors of posttrauma symptom trajectories – including trajectories of PTSS tied to the focal traumatic event (p. 6). 

Second, we return to this point in the Discussion when interpreting our results within the existing literature:

“An additional explanation for the lack of a consistently low PTSS trajectory is that, because we followed an existing cohort, we were able to capture the psychological responses of disaster survivors who otherwise might unlikely to participate postdisaster research studies, such as those who were displaced or coping with major postdisaster stressors. In essence, our results could indicate selection bias in prior trajectory studies, but opposite that posited by Galatzer-Levy and colleagues’ review (7). It is also possible that the review’s finding that resilience was less common in prospective, versus longitudinal, studies could have been due to the fact that mental health symptoms included in the former were not event-specific. Analyzing pre-to-posttrauma trajectories of mental health symptoms not tied to an event (e.g., major depression, non-specific psychological distress) alongside posttrauma trajectories of PTSS in the same prospective cohort would provide greater insight into this issue” (p. 34).

When working on this revision, we also noted that there have been a few studies that, like ours, did not document a trajectory of consistently low PTSS symptoms and that had modal trajectories of initially moderate symptoms that decreased over time. These included a study of sexual assault survivors (Steenkamp et al., 2012 in Journal of Traumatic Stress) and Palestinians exposed to chronic political violence (Hobfoll et al., 2011 in Social Science & Medicine). We therefore have added a statement that, for certain population and under certain conditions, what is commonly referred to as “resilience” might not be the modal response to trauma:

“The lack of what some might term a “resilient” trajectory could be a function of the sample characteristics – specifically, that all participants were female, parents, and low-income at baseline, and most were Black, characteristics that prior research has shown to increase risk for postdisaster psychiatric adversity (Goldmann & Galea, 2015). This speaks to the value of conducting posttrauma trajectory studies within at-risk groups for greater insight into the circumstances under which resilience, defined as a trajectory of consistently low symptoms, is and is not the modal response to trauma. Notably, there have been other prior studies that also did not find a consistently low PTSS trajectory, and like our study, had modal trajectories defined by recovery from initially moderate symptoms. These studies included sexual assault survivors (Steenkamp et al., 2012) and Palestinians exposed to chronic political violence (Hobfoll et al., 2011), further indicating that what is commonly referred to as resilience might be limited in certain subpopulations and/or those facing especially severe trauma” (pp. 33-34, specific references added for clarity).

7) One could argue that the Moderate-Decreasing trajectory is just a variation of a resilient trajectory. Even if you were resilient directly after a disaster, your postdisaster responses (although not on a level of a mental disorder) slowly diminish in the course of time. For this argument one needs to know the mean score of the IES. For the moderate-decreasing trajectory this mean score was at the start of the study 1,22. That implies a sum score of nearly 27 on the IES-22. There are no specific cut-off score for this inventory, but generally scores higher than 24-33 are considered as of either concern or a sign of disorder. So, the term Moderate-Decreasing is justified, but nevertheless this is an issue for the discussion.

This comment was also very thought-provoking. As this Reviewer is likely aware, the term “resilience” has had a few definitions in the literature (e.g., a trait, an outcome, a trajectory) and the trajectory literature seems to have landed on a definition of “consistently low symptoms.” While we agree that the Moderate-Decreasing trajectory might represent some variant of resilience, we opted not to evoke this terminology to further “muddy the waters.” We instead named the trajectories based descriptive data – specifically, alignment between mean PTSS scores and item ratings on the IES-R, and statistical significance of growth terms (p. 23). 

As mentioned in our response to the Editor’s Comment #11, we have used mean IES-R scores to abide by the feedback that we received from one of the scale authors, Daniel S. Weiss, Ph.D. We have pasted text from a document that he provided us at the bottom of the revision letter, supporting the use of mean scores. It is our preference to follow Dr. Weiss’s guidance; however, if the Editor and/or Reviewers would like us to also provide summed scores, we are willing to do so. 

8) The authors should pay more attention in their discussion section to practical implications of their study. They only mention them in a couple of separate sentences (notably, the last sentence of the article).

We agree that our original draft lacked attention to the practical implications of our work. In the revised version, we have added two paragraphs about the study implications for disaster preparedness and response (pp. 35-36).

9) In general, this is a fine study, but the discussion could use more academic reflection. What are theoretical implications of this study? Are there alternative interpretations and explanations? The discussion could go beyond a sober description of the results and implications for future research.

We appreciate this comment in that it pushed us to more thoroughly and meaningfully interpret our findings and situate them within the extant literature. We have made efforts to do so throughout the discussion, for example: interpreting why perceived social support might have reduced to non-significance in multivariable models and how our findings might align with either social causation or social selection processes (p. 32); speculating upon why predisaster physical health conditions or problems might influence trajectory membership by increasing disaster-related exposure (pp. 32-33); drawing on other research to suggest that what is often termed “resilience” might not be the modal response to trauma in certain subpopulations and/or under certain conditions (pp. 33-34); and theorizing that the lower rates of resilience in prior prospective, relative to longitudinal (i.e., posttrauma-only), studies could have been due to their means of symptom assessment and suggesting that our findings provide evidence that longitudinal studies might be overestimating resilience (p. 34). As mentioned previously, we also expanded our discussion of the practical implications of our work for disaster preparedness and response (pp. 35-36). 

Again, we appreciate the opportunity to revise the manuscript and the thoughtful reviews. We believe that, by making the suggested changes, we have improved the quality of the manuscript and we hope that you agree. Thank you in advance for consideration of our work.

Sincerely,

[AUTHORS]

Appendix. Text from document “Using the Impact of Events Score-Revised (IES-R)” by Daniel S. Weiss

Cutting scores, cut-offs, and categorical uses

There are no "cut-off" points for the IES-R, nor are they envisioned or appropriate, despite analyses that present them (e.g., Asukai et al., 2002). The IES-R is intended to give an assessment of symptomatic status over the last 7 days with respect to the 3 domains of PTSD symptoms stemming from exposure to a traumatic stressor. Neither the IES-R, nor the original IES for that matter, was intended to be used as a proxy for a diagnosis of PTSD, and with the very well-developed stable of clinical interviews that were designed to provide diagnoses (Weiss, 2004b), the only reasons to use the IES-R in this fashion is either a misunderstanding of its goals or a choice not to expend the resources (time, funds, good will) to obtain a valid diagnosis. 

This issue is neither new nor confined to symptom measures. Over 30 years ago Rotter (1975) attempted to persuade and cajole researchers interested in the construct of internal-external locus of control not to conceptualize it as a categorical variable, nor to use it that way. The number of “diabetics” in the United States increased dramatically on 1 July 1997, despite any important clinical changes in those who became diabetic. What did change was the official cut-off score promulgated by the American Diabetes Association. For a fasting plasma glucose test, the criterion went from 140 mg per dL (7.8 mmol per L) to 126 mg per dL (7.0 mmol per L). Thus, an estimated 2 million people went to bed on Monday night not being diabetic and on Tuesday morning had become diabetic without any change in their clinical status (Diabetes Monitor, 2005). The use of a classification in and of itself is not necessarily problematic unless it is reified, which it typically is. A short-hand technique for communicating information becomes understood as conveying substantive qualitatively different clinical status. In reality, there are few such situations: pregnant, infected with the HIV virus, dead, boy or girl/male or female. However, even with the last categorization, careful observation has revealed a range of anomalies in which even gender is unclear. The point is that classifying gives the appearance of greater knowledge and understanding than is actually present. My bias is to avoid it if possible.

With respect to the IES-R, there are even more substantive issues that weight against even attempting to set a cut-off score. One of these is the time elapsed since the traumatic event. Early in the course of reaction to traumatic stress, the level of symptoms on the IES-R may suggest the presence of PTSD but distinguishing the normal course of response to trauma from PTSD is a difficult issue at five weeks or two months, regardless of the one month criterion in the DSM. A review of conjugal bereavement (Windholz, Marmar, & Horowitz, 1985) suggested that six months was not out of the ordinary for a period of time during which recovery from the loss. Thus, acute PTSD and chronic PTSD might well require different cuts, if one were to attempt to select them. A second of these is the severity of the traumatic event, all other things being equal, the more severe the higher the symptoms. A third issue is reactions accompanying exposure—both peritraumatic emotionality (Brunet et al., 2001) and peritraumatic dissociation (see Ozer, Best, Lipsey, & Weiss, 2003) may well moderate symptoms and symptom report, in a way that would ultimately affect diagnosis.

Most important, however, is the impact of the base rate of stress reactions in the sample being studied (firefighters versus women who have been beaten during a sexual assault) and used to determine a fixed cut-off. Indeed, in presenting an update on the CAPS, Weathers and colleagues (2001) carefully and systematically describe the need for a variety of decision rules (which are functionally equivalent to a cut-off score), to make a diagnosis of PTSD. They explicitly consider the choice of cut-off in light of the types of errors different values will produce, minimizing or maximizing false positives or false negatives. It has been well known for over five decades (Meehl & Rosen, 1955) that the base rate of the phenomenon can have a sizeable impact on the validity of any cut score. Thus, it is simply inappropriate to require or to attempt to set any cut-off that will universally apply, in which having cut-offs really serves no useful function. 

The choice of the anchor points and the utilization of a mean score, rather than a sum (not universally followed in the literature), was an explicit decision to aid users in interpreting scores. For example, if an individual’s score or a group’s mean on the Intrusion subscale was 1.89, that would indicate that for intrusion, for this person (group), in the last week their distress from intrusive symptoms was close to, but not quite moderate. For individuals similar statements regarding the other two subscales can be made. For groups, using the SD will help immensely in making the pattern of scores meaningful. This ability is consistent with the goal that the IES-R set for itself.

---

## [Decision Letter · Decision Letter 1]

18 Sep 2020

Predisaster predictors of posttraumatic stress symptom trajectories: An analysis of low-income women in the aftermath of Hurricane Katrina

PONE-D-20-05121R1

Dear Dr. Lowe,

We’re pleased to inform you that your manuscript has been judged scientifically suitable for publication and will be formally accepted for publication once it meets all outstanding technical requirements.

Congratulations and kind regards,

Peter G van der Velden, Ph.D.

Academic Editor

PLOS ONE

Additional Editor Comments (optional):

Reviewers' comments:

Reviewer's Responses to Questions

**Comments to the Author**

1. If the authors have adequately addressed your comments raised in a previous round of review and you feel that this manuscript is now acceptable for publication, you may indicate that here to bypass the “Comments to the Author” section, enter your conflict of interest statement in the “Confidential to Editor” section, and submit your "Accept" recommendation.

Reviewer #1: All comments have been addressed

Reviewer #2: All comments have been addressed

2. Is the manuscript technically sound, and do the data support the conclusions?

Reviewer #1: (No Response)

Reviewer #2: Yes

3. Has the statistical analysis been performed appropriately and rigorously? 

Reviewer #1: (No Response)

Reviewer #2: Yes

4. Have the authors made all data underlying the findings in their manuscript fully available?

Reviewer #1: (No Response)

Reviewer #2: Yes

5. Is the manuscript presented in an intelligible fashion and written in standard English?

Reviewer #1: (No Response)

Reviewer #2: Yes

6. Review Comments to the Author

Reviewer #1: (No Response)

Reviewer #2: (No Response)

7. PLOS authors have the option to publish the peer review history of their article (what does this mean?). If published, this will include your full peer review and any attached files.

Reviewer #1: No

Reviewer #2: No

---

## [Editor Report · Acceptance letter]

30 Sep 2020

PONE-D-20-05121R1 

Predisaster predictors of posttraumatic stress symptom trajectories: An analysis of low-income women in the aftermath of Hurricane Katrina 

Dear Dr. Lowe:

I'm pleased to inform you that your manuscript has been deemed suitable for publication in PLOS ONE. Congratulations! Your manuscript is now with our production department. 

Kind regards, 

on behalf of

Dr. Peter G van der Velden 

Academic Editor

PLOS ONE